# Disrupted temperature-sleep coupling mechanism in a Dravet syndrome mouse model

Saja Fadila[1,2,8,9], Georgii Krivoshein [3], Hala Majadly[2,4], Anat Mavashov[2,4], Shahak Ranen[1,2], Marina Brusel[2], Iria G. Dopeso-Reyes[5], Bertrand Beucher[5,6], Eric J. Kremer [5], Else A. Tolner[3,7] & Moran Rubinstein [1,2,4] ✉

Dravet syndrome (DS) is associated with epilepsy, developmental delays, thermal dysregulation, and sleep disturbances. While seizures have been linked to hippocampal dysfunction, what drives sleep disturbances and thermal dysregulation is poorly understood. Using DS mice (Scn1a[A1783V]), we identified a link between sleep and thermoregulation. We found that DS mice exhibited lower core body temperature. Next, using electrocorticography, local field potential recordings, and core temperature monitoring, we showed that DS mice exhibited a lack of core temperature change during the transition from waking to non-rapid eye movement sleep. This is in contrast to wild-type (WT) mice, in which sleep onset coincided with a temperature drop. Additionally, warmth promoted sleep in WT, but not in DS mice. Vector-mediated expression of SCN1A or chemogenetic stimulation of the anterior hypothalamus restored the warmth-induced somnogenesis in DS mice. These findings highlight a connection between sleep and thermal dysregulation in DS, implicating altered neuronal activity of the hypothalamus.

Dravet syndrome (DS) is a developmental and epileptic encephalopathy caused by loss-of-function mutations in the *SCN1A* gene, encoding the α1 subunit of the voltage-gated sodium channel, $Na_V1.1$[1]. In addition to severe epilepsy, developmental delay and cognitive impairment, autonomic dysfunction and sleep problems are also common in patients with DS[2,3]. Autonomic dysfunctions include temperature dysregulation, diaphoresis, and intolerance to cold[4,5]. Sleep problems manifest as nocturnal seizures, trouble falling and staying asleep, and drowsiness[6-9].

Mouse models of DS (DS mice) recapitulate many of the multi-faceted human phenotypes[10-12], including impaired autonomic dysfunction, deficits in respiration, thermoregulation[13-15], circadian dysfunction, and abnormal sleep architecture[16,17]. Among the many existing models, DS mice harboring the missense A1783V *Scn1a* mutation exhibit severe epilepsy and various DS-associated behavioral phenotypes, including spatial memory and motor deficits[12,18-21]. Sleep properties or thermoregulation have not been investigated in this DS model.

Despite their distinct phenotypic manifestations, sleep and thermoregulation are closely associated. In rodents and humans, increased ambient temperature promotes sleep initiation, the onset of non-rapid eye movement (NREM) sleep coincides with a reduction in core temperature, abnormally elevated body temperatures are linked to abnormal sleep homeostasis, and sleep deprivation leads to elevated

[1]Department of Human Molecular Genetics and Biochemistry, Gray Faculty of Medical & Health Sciences, Tel Aviv University, Tel Aviv, Israel. [2]Goldschleger Eye Research Institute, Gray Faculty of Medical & Health Sciences, Tel Aviv University, Tel Aviv, Israel. [3]Department of Human Genetics, Leiden University Medical Center, Leiden, The Netherlands. [4]Sagol School of Neuroscience, Tel Aviv University, Tel Aviv, Israel. [5]Institut de Génétique Moléculaire de Montpellier, Université de Montpellier, CNRS, Montpellier, France. [6]BCM, Univ. Montpellier, CNRS, Inserm, Montpellier, France. [7]Department of Neurology, Leiden University Medical Center, Leiden, The Netherlands. [8]Present address: MRC Laboratory of Molecular Biology, Francis Crick Avenue, Cambridge, UK. [9]Deceased: Saja Fadila ✉e-mail: moranrub@tauex.tau.ac.il

body and brain temperatures[22–25]. The close ties between these two homeostatic physiological responses require a well-coordinated and regulated brain function, especially during the transition states. Indeed, complex neural networks centered in the hypothalamus govern the transitions from wakefulness to sleep and the temperature shifts during these transitions[23,25,26]. However, less is known about the features and mechanisms underlying sleep and temperature transitions in the context of DS. Identifying the neuronal basis of these deficits, as well as developing therapeutic strategies, could improve sleep quality and temperature homeostasis in numerous pathologies.

Here, we demonstrate that sleep and thermoregulation are dysregulated in DS mice ($Scn1a^{A1783V}$). We show deficits in thermal adaptation, a lack of temperature reduction that is normally seen in the transition from wake to NREM sleep, and decreased sleepiness in response to warmth. Moreover, we find that canine adenovirus type 2(CAV2)-mediated $SCN1A$ delivery to, or chemogenetic activation of, anterior hypothalamic neurons rectifies sleepiness induced by a warm temperature. Together, these findings underscore a critical effect of the dysregulated activity of the anterior hypothalamus in the disturbed sleep and thermoregulation phenotype in DS.

## Results

### Dysregulation of body temperature of DS mice in changing ambient temperatures

The ability to stabilize core temperature was tested in DS mice at postnatal day (P) 21-25. At this age, all the mice experience spontaneous seizures[12,18,27]. Interestingly, we noted a lower baseline temperature in DS mice compared to that of wild-type (WT) littermates (Fig. 1a, b). To examine the response to a changing ambient temperature, mice were transferred to a beaker on a heated pad set to 30 °C for 30 min (yielding a moderate warm ambient temperature of 26 °C) and back to RT for another 15 min. Despite starting from a lower temperature, at the end of the 30 min period, the core temperature of WT and DS mice was similar. Yet, upon return to RT, DS mice returned to a lower temperature than that of WT mice (Fig. 1c–e and Supplementary Fig. 1). Of note, while both WT and DS mice reached a temperature that was lower compared to their initial core temperature, the overall temperature of DS mice 15 min after recovery at RT was again lower than that of WT animals (Fig. 1e).

Febrile seizures at relatively low temperatures are a hallmark of $Scn1a^{A1783V}$ mice during their 4th week of life[10,27,28], which guided our decision to use a moderate heat stimulus. Despite that, ~ 25% of the tested DS mice experienced seizures, which led to the termination of the test and exclusion from further analyses (Supplementary Fig. 1).

Thermoregulation is an important step in maintaining homeostasis, especially in preparation for sleep, while nesting contributes to preserving body temperature during sleep[23,29]. When placed overnight with nesting material, juvenile WT mice built complex nests. By contrast, 75% of DS mice did not create nests, and those that did built less complex nests (Fig. 1f–h), demonstrating a reduced ability to perform this innate behavior, which serves as a primary step in sleep preparation. Nevertheless, nesting is a multifactorial behavior that can also be influenced by DS-associated comorbidities, including motor or social deficits, which may contribute to this phenotype.

Together, these data show that DS mice have a lower baseline temperature and display a reduced ability, or possibly a reduced drive, to build a functional nest, which may disturb their preparation for sleep.

### Temperature dysregulation in DS mice during wake to NREM sleep transitions

While sleep deficits occur in DS mice[16,17], an association with temperature dysregulation has not been reported. To explore the link between these physiological processes, we focused on the transition from wake to NREM sleep. This transition is normally associated with

reduced core temperature, and the drop in temperature at NREM onset is important for effective sleep[23,25,29]. To study this transition here, we performed 2–4 h of video electrocorticography/electromyography (ECoG/EMG) recordings in conjunction with continuous measurements of core temperature in WT and DS mice during their 4th week of life (Fig. 2a, b). To identify the transitions to NREM, two investigators analyzed the vigilance states independently, basing their classification on the ECoG and EMG signals, as well as video analysis. Only epochs that were mutually agreed upon were considered for further analysis. Quantification of the average duration of wakefulness and NREM within these short-term recordings revealed a similar duration of wakefulness and NREM sleep in WT and DS, with at least one transition from wake to NREM (Supplementary Fig. 2). Next, we examined the power spectral densities (PSD) of the ECoG during the transition from wakefulness and NREM. The PSD of wakefulness was calculated two minutes prior to the transition to NREM, and the PSD of NREM was measured on the first minute into NREM. In WT mice, a comparison of the PSDs of these states showed that the transition to NREM was accompanied by a clear increase in ECoG delta power. Thus, in WT mice the contribution of delta power to the overall ECoG was larger during NREM sleep than waking . Conversely, in the DS mice, the delta ratio was similar during wake and NREM (Fig. 2c, d). Examination of the core temperature during these transitions to NREM revealed that in WT mice, transitions to NREM were typically (65%) associated with a reduction in body temperature. In contrast, in DS mice, the core temperature remained, on average, stable during transitions, and fewer (21%) transitions were accompanied by a reduction in temperature (Fig. 2e, f).

The anterior hypothalamus is involved in temperature control and the regulation of sleep-wake cycles, also through its influence on the delta frequency band[25,30,31].

To examine features of hypothalamic delta activity during the transition to NREM, we implanted depth electrodes in the anterior hypothalamus and recorded the LFP signal[32] (Supplementary Fig. 3), along with the EMG and continuous core temperature measurement. While the ECoG and LFP signals were overall similar, the contribution of the delta frequency band, which is strongly associated with NREM sleep, tended to be more prominent in the LFP recording (Supplementary Fig. 3d). Of note, as these depth surgeries are more invasive, we used 5-week-old mice. Similar to the ECoG recordings, the hypothalamic depth electrode LFP data showed that in WT mice, transitions from wake to NREM were associated with an increased contribution of the delta frequency band and a reduction in their core temperature. By contrast, in DS mice, the contribution of delta to the overall LFP was similar during wake and NREM, and their temperature was globally stable through these transitions (Fig. 2g–l).

Together, ECoG, LFP, EMG, and core temperature recordings demonstrate that the transition to NREM sleep in DS mice is associated with a diminished increase in delta power at NREM onset and unaltered core temperature.

### Hypothalamic SCN1A expression restores warm-temperature induced sleep promotion in DS mice

A warm ambient temperature (32–36 °C) promotes sleep through a mechanism that involves the activation of heat-sensing neurons in the hypothalamus[29,33]. Given the deficits we observed in DS mice in temperature regulation upon the rise to a moderate ambient temperature of 30 °C, and the sleep transition-related changes in delta power in DS mice, we set out to investigate whether warmth-induced somnogenesis is impaired in DS mice. Depth LFP and EMG activity were recorded in 5-6-week-old WT and DS mice. Recordings were first obtained for one hour at room temperature, followed by an additional hour at elevated ambient temperature achieved by placing the cage on a heating pad set to 36 °C, resulting in a temperature of 29 °C at the bottom of the recording beaker (Fig. 3a–c). Because DS mice at this age

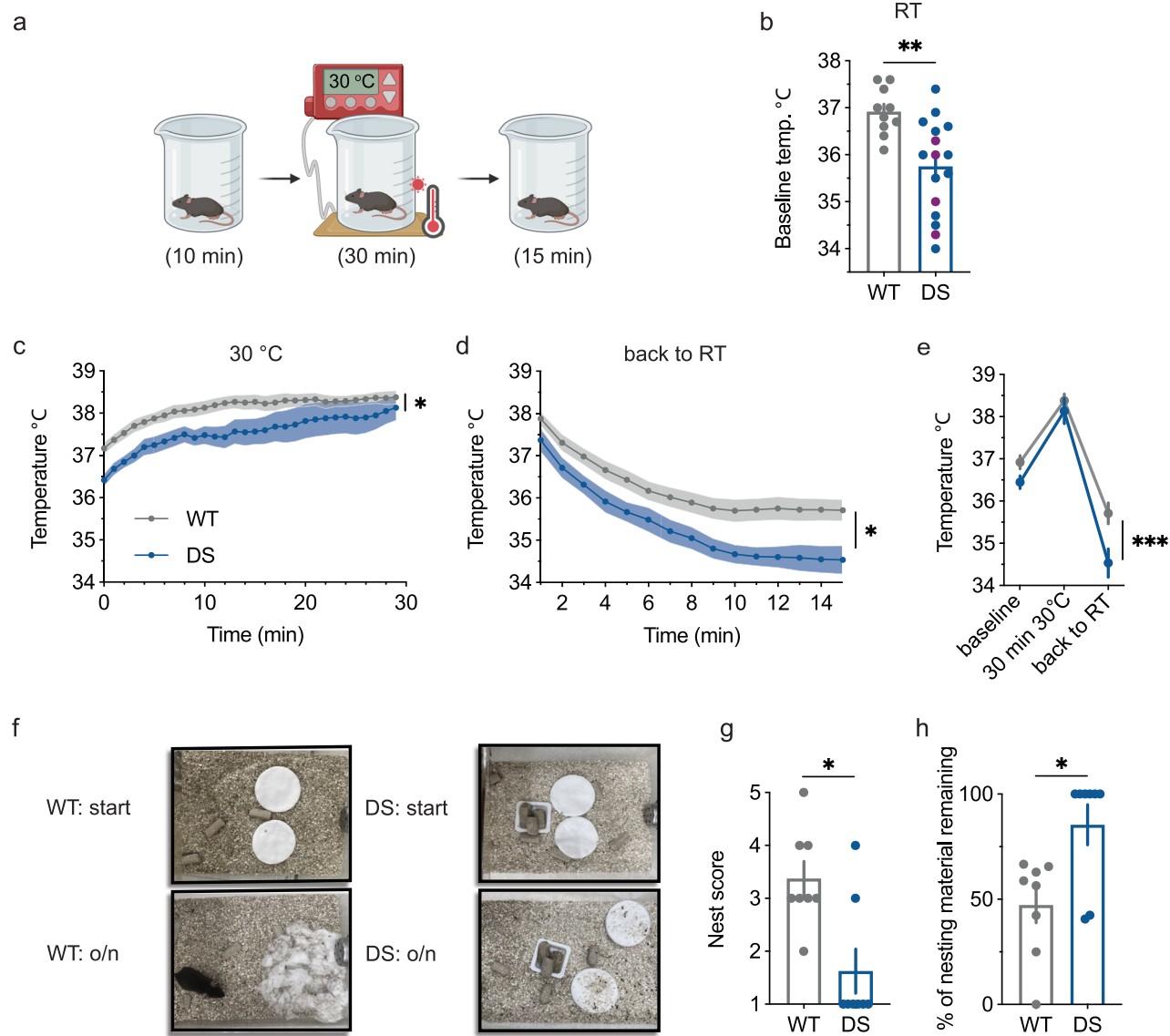

**Fig. 1 | Temperature dysregulation in juvenile DS mice. a** Experimental design. Baseline core temperature was measured starting ~ 30 s after inserting a rectal probe at room temperature (RT), after which the mice were left in the chamber for habituation for 10 min to enable them to get used to the rectal probe. Next, the mice were transferred to a pre-heated beaker placed over a heating pad set to 30 °C (yielding an ambient temperature of 26 °C) with their body temperature being measured for 30 min (**c**), followed by transfer of the mouse to the original chamber at RT for an additional monitoring period of 15 min (**d**). **b** Baseline core temperature at RT is reduced in juvenile (P21 - P25) DS mice. (WT: *n* = 10; DS: *n* = 16; *p* = 0.0012, two-tailed unpaired *t* test). Four out of the 16 DS mice that were tested (denoted in magenta) experienced seizures when placed in the pre-heated beaker (Supplementary Fig. 1). These mice were removed from the chamber and excluded from further experiments. **c** The time course of the increase in core temperature upon placement over a heated pad set to 30 °C. (WT: *n* = 10; DS: *n* = 6. *p* = 0.046, 2-way ANOVA). **d** The time course of the decrease in core temperature upon placement back in RT (*p* = 0.02, 2-way ANOVA). **e** The change in temperature over the course of the experiment. (*p* = 0.008, 2-way ANOVA. The results of Holm-Sidak post hoc analyses between WT and DS are depicted on the graph). **f** Examples of the nesting material at the beginning of the test and after a night (o/n: overnight) spent in the home cage at RT. **g** Nest-building scores show reduced nest-building behavior in DS mice. (WT: *n* = 8; DS: *n* = 8; *p* = 0.014, two-tailed Mann-Whitney test). **h** The percentage of nesting material remaining (*p* = 0.02, two-tailed Mann-Whitney test). The data in panels (**b**–**h**) are presented as mean ± SEM. Individual data points are shown in panels (**b**, **g**, and **h**). See Supplementary Data 1 for full statistical information. Source data are provided as a Source Data file.

show reduced susceptibility to febrile seizures[10,27], we reasoned that a somewhat stronger warming stimulus could be applied compared to the 30 °C condition that was used for the experiments in younger animals (Fig. 1). However, to minimize seizure risk, we avoided the higher temperatures (32–36 °C air temperature) used in previous studies[29,33] making 29 °C a deliberate compromise between physiological relevance and safety in this model.

Vigilance state analyses of the whole recording indicated that the warm temperature increased the time spent in NREM sleep in WT mice (Fig. 3d). Furthermore, during transitions

from wakefulness (2 min prior to NREM onset) to NREM (1 min into NREM), WT mice showed a comparable increase in the relative contribution of delta-band activity to the overall LFP power at both RT and elevated temperature (Fig. 3e). By contrast, warm temperature did not promote NREM sleep in DS mice (Fig. 3d), and the delta ratio during NREM sleep in DS mice was similar at RT and 36 °C (Fig. 3f).

Of note, none of the tested DS mice had a thermally-induced seizure during these recordings. Moreover, the number of interictal spikes in DS mice did not increase for the majority of mice when placed

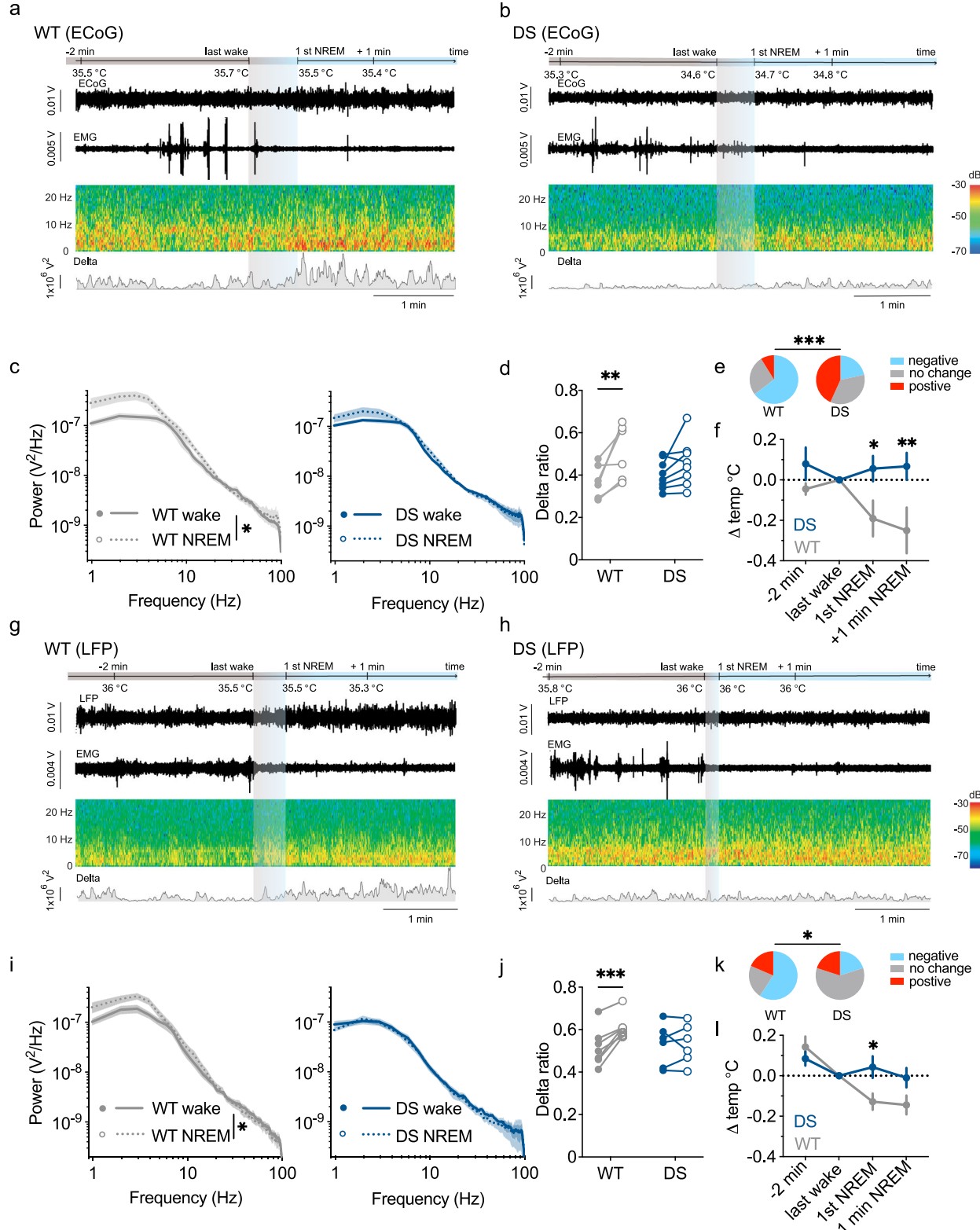

in warm ambient temperature (Supplementary Fig. 5), indicating that increased epileptic activity does not account for the lack of a sleep-promoting effect of a warm environment.

Next, we asked if this is directly related to the DS-associated *Scn1a* loss-of-function mutation, such that increasing Na$_V$1.1 levels in the hypothalamus could ameliorate this phenotype. To address this

possibility, we injected a canine adenovirus type 2 (CAV2) vector harboring an *SCN1A* expression cassette (CAV2-SCN1A)[28] bilaterally into the anterior hypothalamus of WT and DS mice. Hypothalamic LFP and EMG electrodes were implanted 1−2 weeks later. After 3−5 days of recovery, we tested the effect of increased ambient temperature (from RT to 36 °C) on somnogenesis. In WT mice injected with CAV2-SCN1A,

**Fig. 2 | An attenuated increase in delta power and lack of temperature drop during the transition from wake to NREM in DS mice. a, b** Examples showing the combined recording of core temperature, ECoG signal, EMG, sonogram, and the virtual delta showing delta (0.5-4 Hz) power channel in a WT mouse (**a**) and a DS mouse (**b**) during a transition from wakefulness to NREM sleep. **c** ECoG PSD during wakefulness and NREM in WT ($n = 6$) and DS ($n = 8$). 2-way ANOVA for the main effect of sleep, WT: $p = 0.01$, DS: $p = 0.11$. **d** The relative contribution of delta power during the analyzed wake and NREM periods ($p = 0.002$, 2-way ANOVA. The results of Holm-Sidak post hoc analyses are depicted on the graph). **e** The percentage of transitions to NREM sleep that were accompanied by a reduction (negative), unchanged, or an increased core temperature (positive) (WT, $n = 34$ transitions from 6 mice; DS, 37 transitions from 8 mice; $p = 0.0002$, two-sided Fisher's exact test). **f** The average change in core temperature at the indicated time points (2-way ANOVA: $p = 0.037$ for the main effect. The results of Holm-Sidak post hoc analyses are depicted on the graph). **g, h** Examples showing the combined recording of the

core temperature, depth electrode LFP signal, EMG, sonogram, and virtual channel showing delta power in a WT mouse (**g**) and a DS mouse (**h**) during the transition to NREM sleep. **i** LFP PSD during wakefulness and NREM in WT ($n = 7$) and DS ($n = 6$). 2-way ANOVA for the main effect of sleep, WT: $p = 0.013$, DS: $p = 0.7$). **j** The relative contribution of delta power during the analyzed wake and NREM periods ($p = 0.0038$, 2-way ANOVA. The results of Holm-Sidak post hoc analyses are depicted on the graph). **k** The percentage of transitions to NREM sleep that were accompanied by reductions (negative), unchanged, or increased core temperature (positive). ($p = 0.02$, two-sided Fisher's exact test). **l** The average change in core temperature at the indicated time points (2-way ANOVA: $p = 0.04$. The results of Holm-Sidak post hoc analyses are depicted on the graph). The data in (**c**, **e**, **i**, **k**) are presented as mean ± SEM. See Supplementary Fig. 4 for the comparison between genotypes and Supplementary Data 1 for full statistical information. Source data are provided as a Source Data file.

similar to untreated WT mice, the change to 36 °C increased the duration of NREM sleep (Fig. 3j, k). However, a significant increase in the delta contribution at NREM was only observed at RT, likely due to the relatively low number of examined mice in this control cohort (Fig. 3k).

In DS mice, CAV2-SCN1A injection into the hypothalamus had a limited impact on the frequency of interictal spikes (Supplementary Fig. 5). This is in contrast to the strong ameliorating effect on epileptiform activity following CAV2-SCN1A injection into the hippocampus or the thalamus[28]. Nonetheless, CAV2-SCN1A injection in the hypothalamus of DS mice improved temperature-related sleep promotion as measured by enhanced percentage of NREM, and enhanced delta ratio during NREM at RT or warm temperatures (Fig. 3j, l). These data demonstrate that the impaired somnogenesis effect of ambient warmth in DS mice can be corrected by CAV2-mediated expression of Na$_V$1.1 in the anterior hypothalamus.

To directly control for the effects of the hypothalamic CAV2 injections, a separate cohort of WT and DS mice was injected with a control vector containing a reporter gene CAV2-GFP or with CAV2-SCN1A. The biodistribution was examined in mice following CAV2-GFP injection (WT and DS, $n = 3$ for each genotype), which were also used for behavioral and electrophysiological examinations, as well as in a separate group of three WT mice injected with the control vector CAV2-mCitrine, which were only used for immunohistochemistry. The biodistribution analysis demonstrated expression in excitatory and inhibitory neurons in the hypothalamus and, due to CAV2's retrograde transport properties, also in interconnected brain regions (Fig. 4a, b and Supplementary Figs. 8 and 9).

In the cohort of CAV2-GFP or CAV2-SCN1A injected mice, we also evaluated nest-building behavior. Seven to ten days after vector injection, CAV2-SCN1A did not rescue the impaired nest-building abilities of DS mice (Supplementary Fig. 10). Next, ECoG electrodes were implanted, and the mice were allowed to recover for 2-3 days before we proceeded with ~4 h ECoG and temperature recordings at RT. Also in this cohort of mice, CAV2-SCN1A expression in the hypothalamus restored the increase in delta ratio during the transition to NREM sleep, which was absent in DS mice injected with the control vector (CAV2-GFP, Fig. 4a-e). Of note, these analyses were conducted using cortical ECoG rather than the hypothalamic LFP recordings used for the CAV2-SCN1A experiments shown in Fig. 3, confirming that the correction of sleep-related oscillatory activity by CAV2-SCN1A is also reflected at the cortical level. Core temperature measurements were included in this new cohort of mice to test whether hypothalamic SCN1A expression may also correct the drop in body temperature at NREM onset. However, in contrast to the data from naïve untreated mice shown in Fig. 2, no significant temperature decrease was detected after NREM onset in either WT or DS mice injected hypothalamically with either CAV2-GFP or CAV2-SCN1A. The discrepancy between these results and those presented in Fig. 2 may be attributable to the

different experimental procedures, as the mice depicted in Fig. 2 underwent only one surgical process, for electrode implantation (ECoG at the age of 4 weeks, or LFP at the age of 5 weeks), while the mice shown in Fig. 4 underwent two surgeries, one for vector injection at 4 weeks and a second one for ECoG implantation at 5 weeks.

The warmth-induced somnogenesis experiment was also performed in this cohort of mice, 1–5 days after the ECoG recordings at RT, and confirmed that NREM sleep was promoted in DS mice injected with CAV2-SCN1A, whereas in DS mice injected with the control CAV2-GFP vector, the switch to warm temperature did not induce NREM sleep (Fig. 4f–j).

Together, the data shown in Figs. 3 and 4 show that CAV2-mediated SCN1A expression in the hypothalamus increases the delta contribution to NREM sleep and corrects somnogenesis in a warm environment.

## Chemogenetic activation of hypothalamic neurons restores warm-temperature-induced sleep promotion in DS mice

The prevailing view is that DS-associated *Scn1a* mutations lead to reduced neuronal excitability due to impaired Na$_V$1.1 channel function[34,35]. The hypothalamus comprises a heterogeneous population of excitatory and inhibitory neurons, both expressing comparable levels of *Scn1a* mRNA (Supplementary Fig. 8)[36]. If the primary effect of CAV2-SCN1A is to enhance the activity of transduced neurons, we wondered whether artificially increasing neuronal excitability, through chemogenetic activation, could mimic the effect of exogenous Na$_V$1.1 expression. Specifically, we tested whether chemogenetic activation of hypothalamic neurons could restore temperature-dependent sleep induction, similar to the correction observed following CAV2-SCN1A injections.

CAV2-hM3D-IRES-mCitrine was injected into the anterior hypothalamus of 4-week-old WT or DS mice. CAV2-hM3D-IRES-mCitrine encodes hM3D, a CNO (clozapine N-oxide)-activatable DREADD and an mCitrine expression cassette. 1-2 weeks post-injection, LFP and EMG electrodes were implanted, and the mice were examined 3–5 days later for the sleep-promoting effect of increased ambient temperature (~6 weeks old during the recording). As a control for the CNO injection, mice were first tested following the injection of saline just before the cage was placed on a 36 °C heating pad. A few days later, the experiment was repeated using CNO injection (10 mg/kg) (Fig. 5a, b). This dosage of CNO did not alter locomotor activity, as assessed in an open field test in WT mice without DREADD expression (Supplementary Fig. 12), indicating the absence of a nonspecific sedation effect at this dose[37].

In WT mice, the warm temperature promoted NREM sleep after saline or CNO injection (Fig. 5c–f), similarly to untreated WT mice (Fig. 3d, e). The higher delta ratio during NREM sleep compared to waking was observed in some of the conditions, but not all (Fig. 5f). By contrast, in saline-injected DS mice, similar to untreated DS mice (Fig. 3d, f), or CAV2-GFP injected DS mice (Fig. 4f), the warm

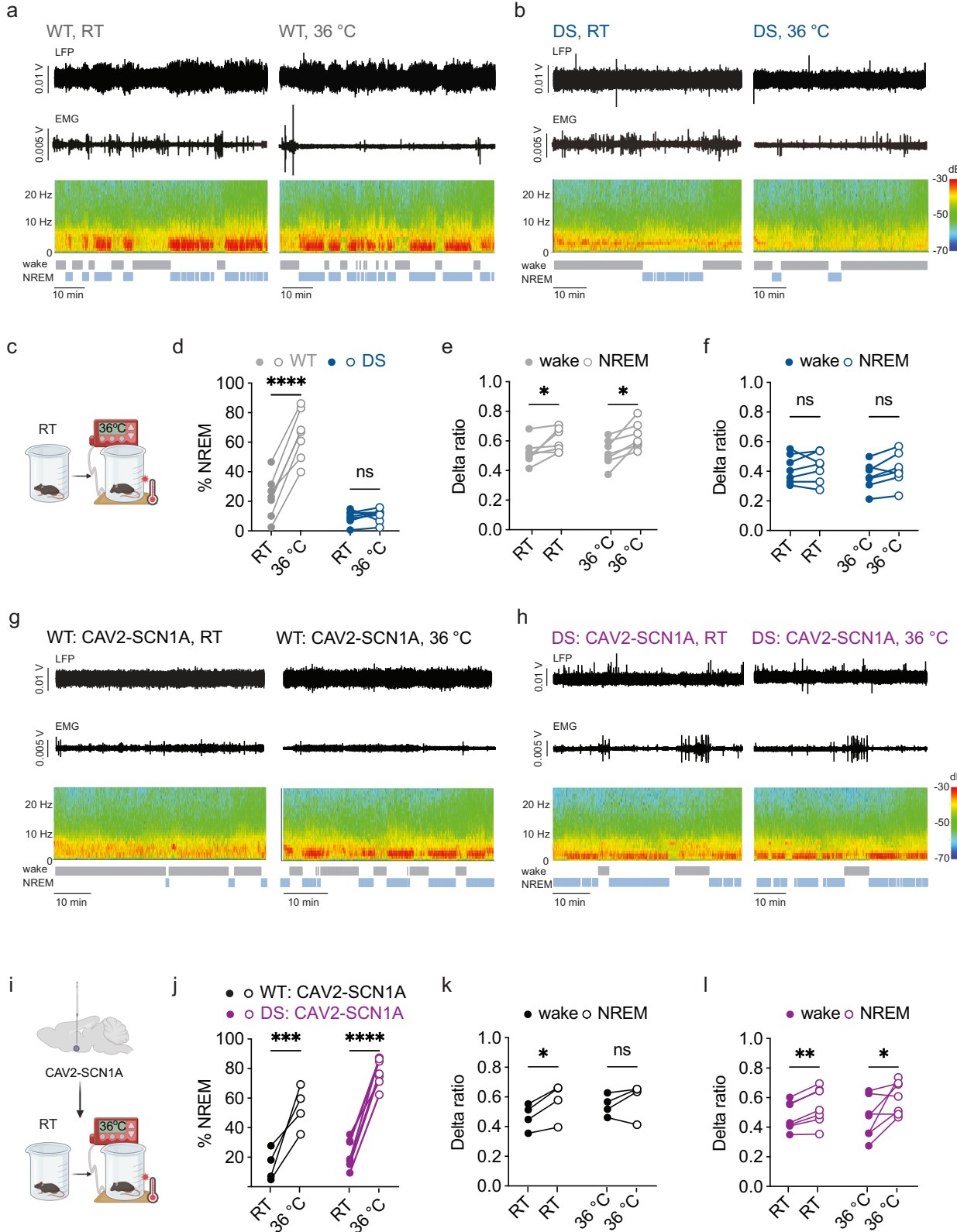

temperature did not promote NREM sleep (Fig. 5g–i). Conversely, in the same DS mice, following CNO administration, the warm temperature promoted NREM sleep, as well as an increase in the contribution of delta power at NREM, but only in warm temperature after CNO injection (Fig. 5h–j). Thus, acute chemogenetic activation of hypothalamic neurons was sufficient to restore warm temperature-induced sleep promotion in DS mice.

## Discussion

Among the multifaceted comorbidities of DS, thermal dysregulation and sleep deficits are significant factors that impact the lives of patients and families. These deficits are also found in DS mice. Here, we used a DS mouse model harboring the missense mutation homologous to the recurrent human pathogenic *SCN1A* A1783V to characterize DS-associated deficits in these linked functionalities. We show that DS

**Fig. 3 | Impaired warmth-induced somnogenesis in DS mice can be corrected by CAV2-mediated SCN1A expression in the anterior hypothalamus. a–f** Warm temperature promotes NREM sleep in untreated WT mice, but not in untreated DS mice. **a, b** Examples of depth LFP, EMG, sonogram and hypnogram in a WT (**a**) and DS (**b**) mice at RT and 36 °C. **c** A depiction of the experimental design. (**d**) Percentage of time spent in NREM at each temperature (WT: $n = 7$; DS: $n = 7$. $p < 0.0001$, 2-way ANOVA. The results of Holm-Sidak post hoc analyses within genotypes are depicted on the graph. WT: %NREM RT vs 36 °C, $p < 0.001$. DS: %NREM RT vs 36 °C, $p = 0.67$. **e** The relative contribution of delta power to wake and NREM sections in WT mice (WT: $n = 7$; $p = 0.007$, 2-way ANOVA. The results of Holm-Sidak post hoc analyses are depicted on the graph). **f** The relative contribution of delta power to wake and NREM sections in DS mice (DS: $n = 7$; $p = 0.16$, 2-way ANOVA). **g–l** Warm temperature promotes sleep in both WT and DS mice following hypothalamic CAV2-SCN1A expression. **g, h** Examples of depth LFP, EMG, sonogram, and hypnogram in a WT: CAV2-SCN1A (**g**) and DS: CAV2-SCN1A (**h**) mice at RT and 36 °C. **i** A depiction of the experimental design. **j** Percent of time spent in NREM at each temperature (WT: CAV2-SCN1A, $n = 4$; DS: CAV2-SCN1A, $n = 7$. $p < 0.0001$, 2-way ANOVA. The results of Holm-Sidak post hoc analyses within each genotype are depicted on the graph. WT: CAV2-SCN1A: %NREM RT vs 36 °C, $p = 0.002$. DS: CAV2-SCN1A: %NREM RT vs 36 °C, $p < 0.0001$. **k** The relative contribution of delta power to wake and NREM sections in WT: CAV2-SCN1A mice ($n = 4$; $p = 0.08$, 2-way ANOVA. The results of Holm-Sidak post hoc analyses are depicted on the graph). **l** The relative contribution of delta power to wake and NREM sections in DS: CAV2-SCN1A mice (DS: $n = 7$; $p = 0.018$, 2-way ANOVA. The results of Holm-Sidak post hoc analyses are depicted on the graph). See Supplementary Data 1 for full statistical information. See Supplementary Fig. 6 for additional statistical comparison between genotypes. Source data are provided as a Source Data file.

mice display altered responses to a changing ambient temperature and impaired nesting abilities. Focusing on the transition to sleep, we identified deficits in temperature reduction at the onset of NREM and impaired warmth-induced somnogenesis. Finally, we demonstrated restoration of the warmth-induced sleep-promoting features by modulation of hypothalamic neuronal activity by either CAV2-mediated expression of *SCN1A* or chemogenetic stimulation of anterior hypothalamic neurons. Together, these findings highlight the involvement of dysregulated neuronal activity in the anterior hypothalamus in these DS-associated non-epileptic comorbidities.

The physiological transitions between wakefulness and sleep require the coordinated function of multiple brain regions to modulate the vigilant state and body temperature[23,25,26]. Sleep initiation involves the activation of cortico-thalamic circuits as well as neuronal pathways between the anterior hypothalamus, particularly the preoptic area, and the basal forebrain[38]. Body temperature regulation involves skin thermosensitive receptors that send signals to the preoptic area of the anterior hypothalamus, as well as the brainstem, which then controls vasodilation and heat regulation[23,39]. Intriguingly, sleep-promoting networks and thermoregulation pathways converge in the preoptic region of the anterior hypothalamus[33,40]. Moreover, this region is also known to modulate delta oscillations during NREM[31,41]. Here, we show in DS mice an average stable temperature during the transition from wake to NREM (Fig. 2), diminished delta contribution increase at NREM onset (Figs. 2–4), and a lack of warmth-induced somnogenesis (Fig. 3). The finding that the latter phenotype could be restored by targeted overexpression of SCN1A or chemogenetic neuronal activation in the hypothalamus (Figs. 3–5) indicates that dysregulated hypothalamic function may contribute to this non-epileptic comorbidity. This is in addition to the reported activation of the anterior hypothalamus in a rat model of DS during heat-induced seizures[42].

What might be the mechanism governing dysfunction of the anterior hypothalamus in DS? DS-associated loss-of-function *Scn1a* mutations hinder the excitability of multiple types of inhibitory neurons throughout the brain[35,43,44], alter the function of glutamatergic excitatory neurons[45–47], and lead to dysfunction of cortical, hippocampal, and thalamic neuronal circuits[19,45,48–52]. Moreover, GABAergic inhibitory neurotransmission is impaired in the suprachiasmatic nucleus of the hypothalamus, the primary site of the circadian clock, and pharmacological enhancement of GABAergic inhibition restored normal function of the circadian rhythm in a DS *Scn1a* KO model[53]. Accordingly, we propose that the deficits in DS mice, including the dysregulation of core temperature during transition to sleep and the reduced sleep-promoting effect of warmth, are related to DS-associated neuronal dysfunction in the anterior hypothalamus.

The preoptic region in the anterior part of the hypothalamus hosts a diverse neuronal population with ~50% inhibitory neurons, and comparable *Scn1a* expression in both excitatory and inhibitory neurons[36,54]. Neurons in the hypothalamus are critical for thermal regulation and sleep promotion through projections to other hypothalamic nuclei and brain regions[25,40]. Optogenetic and chemogenetic stimulation of GABAergic inhibitory neurons in the anterior hypothalamus in wild-type mice reduced the core temperature[33,55–57], and chemogenetic stimulation driven by the activity-dependent cFos promoter (activating both excitatory and inhibitory neurons) in this region promoted sleep[58]. Moreover, preoptic heat-responsive glutamatergic neurons were shown to respond to environmental warmth, promoting sleep and body cooling by activating downstream inhibitory neurons, underscoring the key role of anterior hypothalamic neurons in thermoregulation and sleep[59].

In light of the intricate neuronal population of the hypothalamus, which includes both sleep- and wake-promoting neurons[57], pinpointing the cellular basis of these DS-associated sleep and thermoregulatory deficits remains challenging. This complexity is further illustrated by the presence of hypothalamic neuron types with unique properties, such as wake-promoting glutamatergic parvalbumin-positive neurons[60]. Moreover, since *Scn1a* is expressed in both excitatory and inhibitory hypothalamic neurons[36], identifying the precise cellular mechanisms involved becomes even more difficult. However, the observation that hypothalamic neuronal CAV2-mediated SCN1A expression or chemogenetic stimulation restored the warmth-induced sleep-promoting response suggests that reduced neuronal excitability, resulting from *Scn1a* loss-of-function mutation, is a plausible mechanism for the observed DS-related deficits. Nevertheless, CAV2-mediated expression of SCN1A or hM3D is not restricted to a specific class of neurons. CAV2-SCN1A contains the neuronal-specific enolase (NSE) promoter, and CAV2-hM3D-IRES-mCitrine harbors the CMV (cytomegalovirus early enhancer) promoter. Both promoters lead to transgene expression in excitatory as well as inhibitory neurons[28]. Moreover, with the effective retrograde transport properties characteristic of CAV2[61,62], injection of the CAV2-based vectors to the anterior hypothalamus also transduce neurons in regions that project to the hypothalamus. Thus, it is possible that the anterior hypothalamic injection of CAV2-SCN1A leads to a correction in the activity of multiple types of neurons in this region, as well as additional connecting networks that hub on the anterior hypothalamus, that together contribute to the observed improvement in sleep and thermoregulation in DS mice.

In contrast to other studies that focused on developing therapies for DS[28], the use of CAV2-SCN1A in the context of this study was to gain mechanistic insights and link the observed DS deficit in the contribution of delta to NREM and warmth-induced somnogenesis with the DS-associated *Scn1a* loss-of-function. Moreover, as the number of interictal spikes was similar in untreated mice and mice injected with CAV2-SCN1A, the hypothalamus may not be a critical driver for epileptiform activity or seizure expression in DS. Nesting, which is linked to preoptic hypothalamic activity[63–66], also remained abnormal in DS mice despite CAV2-SCN1A expression in the hypothalamus, indicating that, in the context of DS, additional

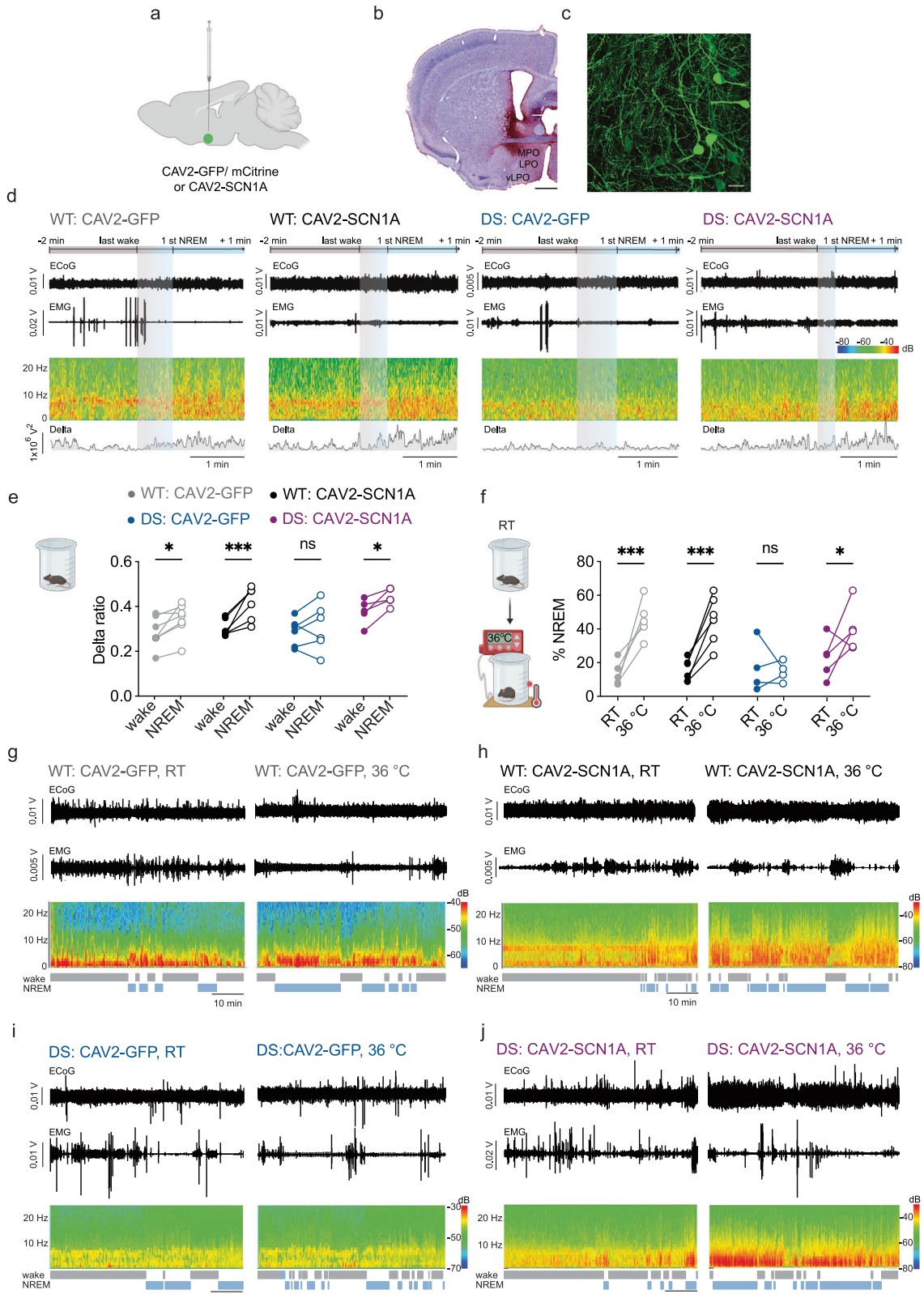

comorbidities or neuronal networks contribute to this phenotype, and impaired nesting may further contribute to sleep temperature dysregulation. Conversely, as CAV2-SCN1A injection into the hippocampus or thalamus corrected the epileptic and cognitive phenotypes of DS[28], it remains to be determined whether CAV2-SCN1A injection to these sites could also correct the sleep and thermoregulation deficits described here.

Previous long-term analyses of sleep patterns in DS mice from the *Scn1a* KO model demonstrated an altered circadian rhythm and fragmented sleep[16,17,53]. Our data from the *Scn1a* A1783V missense model complement these findings by demonstrating the diminished enhancement in the delta band activity at the onset of NREM, which was evident from both ECoG and depth electrode LFP recordings. Increased delta power indicates deep NREM sleep, and neurological

**Fig. 4 | CAV2-SCN1A restores NREM sleep delta ratio and warmth-induced somnogenesis in DS mice. a, b** The control vector CAV2-mCitrine was injected into the hypothalamus. LPO: lateral preoptic area, MPO: medial preoptic area, vLPO: ventrolateral preoptic nucleus Scale bar: 1 mm. **c** mCitrine expression in the vLPO. Scale bar 10 µM. Representative result from six independent experiments. **d** Examples showing the ECoG signal, EMG, sonogram, and the virtual delta depicting delta (0.5-4 Hz) power in a WT: CAV2-GFP, WT: CAV2-SCN1A, DS: CAV2-GFP and DS: CAV2-SCN1A during a transition from wakefulness to NREM sleep. **e** About 2 weeks after viral injection, the mice were recorded for 4 h at RT. The relative contribution of delta power during the analyzed wake and NREM periods (WT: CAV2-GFP, $n = 7$. WT: CAV2-SCN1A, $n = 6$. DS: CAV2-GFP, $n = 6$. DS: CAV2-SCN1A, $n = 5$. $p = 0.023$, 2-way ANOVA. The results of Holm-Sidak post hoc analyses

are depicted on the graph). **f** The effect of warm temperature was tested in the same mice 1-5 days later, at this stage, the mice were recorded for 1 h at RT, and then on a pad heated to 36 °C. The percent of time the mice spent in NREM sleep at each temperature is depicted (WT: CAV2-GFP, $n = 5$. WT: CAV2-SCN1A, $n = 6$. DS: CAV2-GFP, $n = 4$. DS: CAV2-SCN1A, $n = 5$. $p < 0.0001$, 2-way ANOVA. The results of Holm-Sidak post hoc analyses are depicted on the graph. WT: CAV2-GFP %NREM at RT and 36 °C $p = 0.0007$. WT: CAV2-SCN1A %NREM at RT and 36 °C $p = 0.0007$. DS: CAV2-GFP %NREM at RT and 36 °C $p = 0.75$. DS: CAV2-SCN1A %NREM at RT and 36 °C $p = 0.041$). **g–j** Examples of ECoG, EMG, sonogram and hypnogram of WT: CAV2-GFP (**g**), WT: CAV2-SCN1A (**h**), DS: CAV2-GFP (**i**), DS: CAV2-SCN1A (**j**). Source data are provided as a Source Data file.

disorders with a negative impact on sleep, like insomnia and traumatic brain injury, tend to display fragmented sleep with reduced delta contribution during NREM[67–69]. Moreover, delta power decreases with aging in association with impaired sleep features and age-related increased risk for cognitive impairment[70,71]. Thus, it is possible that a failure to enhance delta activity upon sleep initiation in DS contributes to the overall poor sleep quality and cognitive deficits.

Another observation we made was that juvenile DS mice, at the beginning of their fourth week of life, showed a lower baseline core temperature compared to WT mice. This difference could be attributed to altered metabolic homeostasis described in DS[72], involving dysregulation of energy metabolism and mitochondrial function, or to the severe epilepsy typical at this age. Both factors may contribute to the reduced thermoregulatory stability observed across sleep-wake transitions. In the DS model used here, all the mice experience spontaneous seizures during their fourth week of life[12,27]. The lower baseline temperature may therefore be related to post-ictal hypothermia, which has been reported in both human epilepsy patients[73] and DS mice up to several hours after a seizure[74]. In older DS mice at 5-6 weeks of age, when spontaneous seizures become less frequent[12,27], the baseline temperature difference between DS and WT mice was less pronounced, which is consistent with a reduced seizure burden, as well as a reduction in metabolic stress. Although we did not assess metabolic parameters in the present study, this will be an important direction for future investigations to better define the contribution of metabolic mechanisms to thermoregulatory regulation in DS. Nonetheless, sleep and thermoregulation deficits were observed in both juvenile (4-week-old) and older (5-6-week-old) DS mice, demonstrating that the phenomenon is not confined to a single age group or disease stage.

A limitation of the present study is that we did not directly assess how the effects of the DS *Scn1a* mutation and hypothalamic CAV2-mediated SCN1A expression, or alternatively chemogenetic stimulation, influence neuronal firing and excitability in the anterior hypothalamic region. Future studies addressing these questions could provide deeper mechanistic insight into the neuronal basis of sleep and thermoregulation deficits in DS.

Together, these results demonstrate DS-associated impairments in thermoregulation and warmth-induced sleep onset, which can be ameliorated by CAV2-mediated expression of the *SCN1A* gene or chemogenetic enhancement of neuronal excitability in the anterior hypothalamus. The involvement of the hypothalamus in these DS-associated non-epileptic comorbidities adds to our understanding of the mechanisms and brain regions involved in DS.

## Methods
### Study approval
All animal experiments were approved by the Institutional Care and Use Committee of Tel Aviv University.

### Animals
WT and DS mice harboring the global *Scn1a*[A1783V/WT] mutation were generated by crossing conditional floxed *Scn1a*[A1783V/WT] males (The

Jackson Laboratory; stock #026133, C57BL/6 J) with CMV-Cre females (The Jackson Laboratory; stock #006054, C57BL/6 J), as described before[12,27,28]. The mice were kept on the pure C57BL/6 J genetic background. Mice were housed in a standard animal facility at a constant temperature of 22 °C, on a 12 h light/dark cycle, with ad libitum access to food and water. Both male and female WT and DS mice were used in all of the experiments, and the data were combined.

### Temperature regulation in a changing ambient temperature
WT and DS mice at the age of P21-P25 were used (4th week). The mice were handled for 5 min prior to the test. To measure the core temperature of the mice, we used a rectal thermal probe connected to a digital temperature controller (RET-4 probe, connected to TCAT-2DF, Physitemp Instruments, Clifton, NJ, USA). The baseline temperature was measured 30–60 sec after the probe was inserted and continuously for the duration of the experiment using a video recording of the digital temperature controller. The mice were given an additional 10 min to habituate to the rectal probe at room temperature (RT), and the experiment started as the mouse was moved to the test beaker, which was placed on a heated pad set to 30 °C (controlled by a second digital temperature controller, TCAT-2DF, Physitemp Instruments, Clifton, NJ, USA). Under these conditions, the temperature at the bottom of the beaker was 26 °C. We used the moderate heat of 30 °C in DS mice at their fourth week of life due to their high sensitivity to thermally induced seizures[10,27]. Next, the mouse was moved to a different beaker at RT, and the body temperature was measured for another 15 min. This was the protocol for ten WT mice and six DS mice (Fig. 1). Three more WT mice and six more DS mice were only followed when placed over the heated pad, and not during the cooling stage (Supplementary Fig. 1). The core temperature was extracted from the video in one-minute intervals. The test was terminated immediately if DS mice showed any signs of seizures (Supplementary Fig. 1).

### Nest building assay
Two hours before the dark period, mice aged P21-22 (Fig. 1) or P31–33 (Supplementary Fig. 10) were placed in individual cages with wet food and water. The bedding materials were weighed and placed in the middle of the cage. The next morning, the unused nesting materials were weighed. In addition, the nests were given a score on a scale from 1–5, with a score of 1 in case the bedding materials were untouched and a score of 5 if most ( > 95%) of the materials were used[75].

### ECoG surgery
Electrode implantation was performed in mice as described before[12,28,76]. Briefly, mice were injected with ketamine/xylazine (191/4.25 mg/kg) for anesthesia. Carprofen (5 mg/kg) was injected for analgesia prior to the procedure. A midline incision was made above the skull, and two electrodes of fine (130 µm bare diameter; 180 µm coated diameter, cat # 785500 A-M systems, Carlsborg WA, USA) silver wires were placed under the dura at visually identifiable locations over the somatosensory cortex. A reference electrode was placed over the

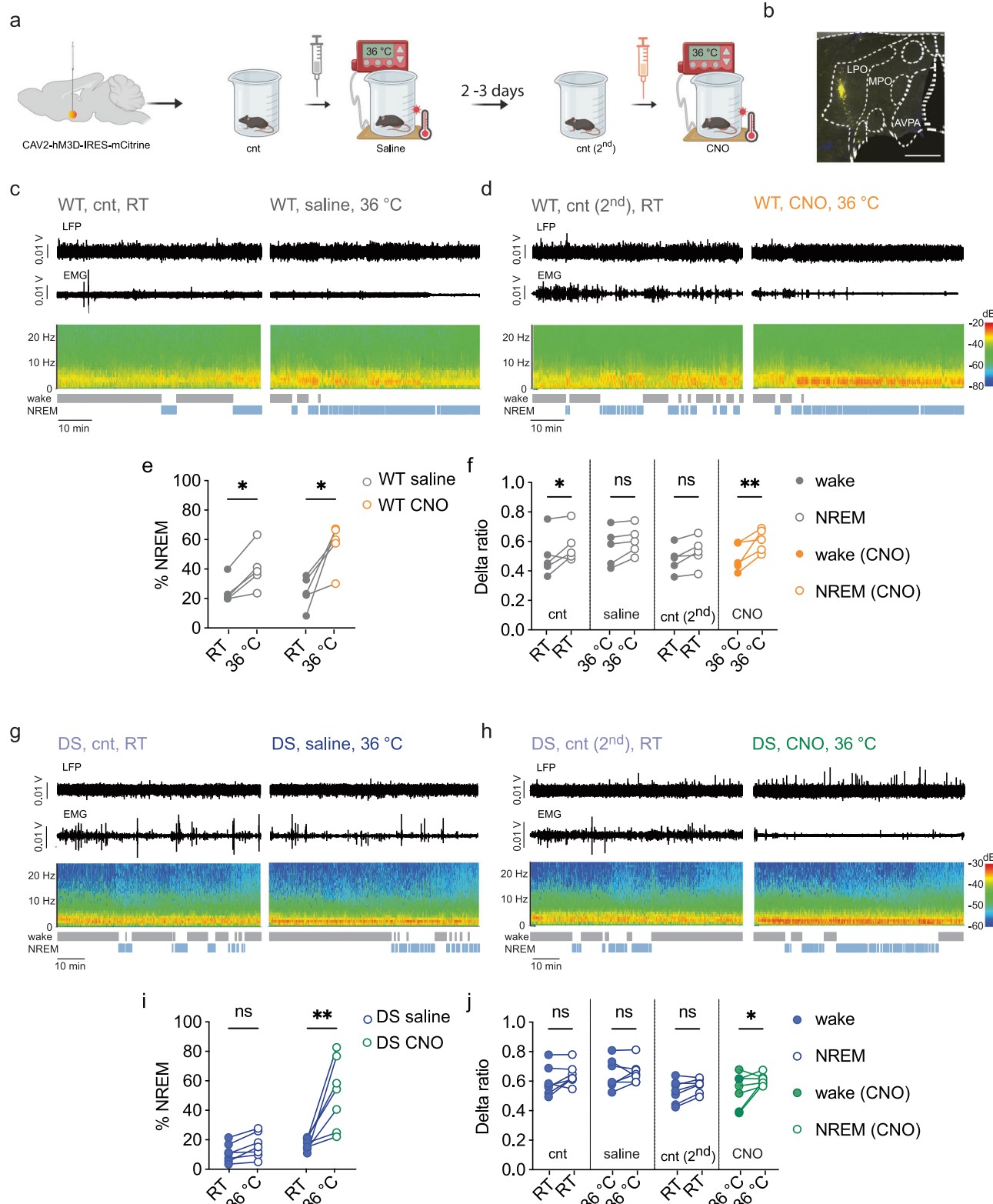

cerebellum. Two electrodes with ball tips were used for EMG, which was placed above the neck musculature on one side, and a ground electrode, which was placed under the skin over the shoulder on the other side. The millmax connector was secured with dental cement, and the skin was closed with sutures. For the ECoG recordings in Fig. 2, we used mice at their 4th week of life. For the ECoG recordings in Fig. 4, the mice were 5-6 weeks old. The mice were given 2-3 days for recovery after surgery before recording.

**LFP depth electrode surgery**

Five to six week old mice were anesthetized as described for the ECoG surgery and were mounted on a stereotaxic frame (Ultra Precise Stereotaxic Instruments, Stoelting, Wood Dale, IL, USA). Fine silver wire electrodes (same type as used for ECoG) were inserted into the lateral hypothalamus (AP: + 1 mm, ML: ± 1 mm, DV:-5.2 mm). A reference electrode was placed over the cerebellum. An EMG electrode was placed above the neck musculature on one side, and a ground electrode was

**Fig. 5 | Correction of the reduced warmth-induced somnogenesis in DS mice by acute chemogenetic-mediated activation of the hypothalamus. a** A schematic representation of the experimental design. Saline injection was given right before the mouse was placed on the heated pad. Two to three days later, the experiment was repeated, and CNO injection was administered just before the mouse was placed on the heated pad. The first 15 min following CNO injection were excluded from the analyses. **b** The expression of mCitrine in the anterior hypothalamus. LPO: lateral preoptic area, MPO: medial preoptic area, AVPV: anteroventral periventricular nucleus. Scale bar: 0.5 mm. **c, d** Examples of depth LFP, EMG, sonogram and hypnogram in the same WT mouse at RT and at 36 °C after saline injection (**c**), or at RT and 36 °C after CNO (10 mg/kg) injection (**d**). **e** Percent of time spent in NREM at each temperature after saline or CNO injection (WT: $n = 5$; $p = 0.005$, 2-way ANOVA. The results of Holm-Sidak post hoc analyses are depicted on the graph). **f** The

relative contribution of delta power to wake and NREM sections in WT mice at the different parts of the experiment (temperature and saline vs. CNO) (WT: $n = 5$; $p = 0.012$, 2-way ANOVA. The results of Holm-Sidak post hoc analyses are depicted on the graph). **g, h** Examples of depth LFP, EMG, sonogram, and hypnogram in the same DS mouse at RT and at 36 °C after saline injection, or at RT and 36 °C after CNO (10 mg/kg) injection. **i** Percent of time spent in NREM at each temperature after saline or CNO injection (DS: $n = 7$; $p = 0.018$, 2-way ANOVA. The results of Holm-Sidak post hoc analyses are depicted on the graph). **j** The relative contribution of delta power to wake and NREM sections in DS mice at the different parts of the experiment (temperature and saline vs. CNO) (DS: $n = 7$; $p = 0.15$, 2-way ANOVA. The results of Holm-Sidak post hoc analyses are depicted on the graph). Source data are provided as a Source Data file.

placed under the skin over the shoulder on the other side. In a subset of animals, the electrodes were coated with DiI (1,1-dioctadecyl-3,3,3,3-tetramethylindocarbocyanine perchlorate, 10% in ethanol) to verify their location. In others, after the recordings, the brains were extracted to 4% PFA and sectioned to verify the electrode location. The mice were given 3–5 days for recovery after surgery before recording.

### Viral injection into the hypothalamus
WT and DS mice, at the age of P22-26, were injected with CAV2-NSE-SCN1A, CAV2-NSE-GFP, CAV2-NSE-mCitrine (CAV2-SCN1A, CAV2-GFP or CAV2-mCitrine[28]), or CAV2-hM3D-IRES-mCitrine (Plateforme de Vectorologie de Montpellier, IGMM Montpellier, France). The mice were anesthetized using ketamine/xylazine (191/4.25 mg/kg), and carprofen (5 mg/kg) was used for analgesia. The mice were placed in a stereotaxic device (Ultra Precise Stereotaxic Instruments, Stoelting, Wood Dale, IL, USA). A midline incision was made above the skull, and holes were made using a 25 G needle at the site of injection. We used the following coordinates for: AP + 0.9 mm; ML ± 0.9 mm; DV − 5.2 mm. The injections were done at a rate of 100 nl/min. One microliter containing $1 \times 10^9$ physical particles was injected (0.5 μl on each side) using a 1 μl beveled needle Hamilton syringe (7000 series, Hamilton, Reno, NV, USA). The needle was kept in place for 10 minutes following injection on each side. Then, the needle was retracted slowly to prevent leakage of the vector. The mice were allowed to recover from the surgery and were kept in isolation for 1-week following the injection. Seven to fourteen days later, the same mice underwent surgery to implant the LFP depth electrode (Figs. 3, 5), or ECoG (Fig. 4) as described above.

For the histological analyses, the mice were perfused and sliced as described before[28].

### Video-ECoG /LFP- and core temperature recordings
Video-ECoG/LFP recordings were performed in a separate room during the light period (8 am-6 pm) for 2-5 h per mouse in a padded Plexiglas box (Figs. 2, 4d) or a 2-liter beaker (Figs. 3, 4g–j, 5) with wet food provided. The mice were connected to a T8 Headstage (gain X100 Triangle BioSystems, Durham, NC, USA), using PowerLab 8/35 acquisition hardware and LabChart 8 software for analyses (ADInstruments, Colorado Springs, CO, USA). The sampling rate was 1 kHz with a notch filter at 50 Hz, and a band-pass filter of 0.5-100 Hz. For the recordings depicted in Figs. 2, 4d, the core temperature was continuously measured using a rectal probe connected to a digital temperature controller (RET-4 probe, connected to TCAT-2DF, Physitemp Instruments, Clifton, NJ, USA), of which the digital display was monitored continuously by video. The recordings were long enough, and the conditions were suitable for the vast majority of the mice (except for one DS mouse, which was excluded from the analysis) to reach NREM sleep, but not REM sleep.

### Depth electrode LFP or ECoG recordings in a changing ambient temperature
LFP recordings were performed as indicated above, with the exception of a subset of data that are presented in Fig. 5, which were performed

using Neuralynx Digital Lynx data acquisition system (Neuralynx, Inc., Bozeman, MT, USA) using the same configurations as those recorded using PowerLab 8/35. To test the effect of temperature on sleep promotion, the mice were recorded in two blocks: the first one at room temperature for one hour, after which the recording chamber was moved and placed over a heated pad set to 36 °C (TCAT-2DF, Physitemp Instruments, Clifton, NJ, USA), and a second block was recorded for another hour. Under these conditions, the air at the bottom of the beaker was at ~ 29 °C. Note that for these experiments, since the DS mice were older than five weeks, a higher ambient temperature of 36 °C was used (and not the 30 °C used for the younger DS mice in Fig. 1), given reduced sensitivity to thermally induced seizures at this age. For the experiments described in Fig. 5, immediately prior to the placement over the heated pad, the mice were given an IP saline injection, and when the experiment was repeated 3-4 days later, an IP injection with 10 mg/kg clozapine-N-oxide (CNO, Angene, London, UK). We analyzed the entire recording after saline injection, but excluded the first 15 min following CNO injection to allow for CNO absorption[77,78]. Locomotor activity of WT mice, without DREADD expression, was not affected by 10 mg/kg CNO. Specifically, 15 min after CNO injection, locomotor activity of the mice was monitored for 10 min in an open field of 50 by 50 cm (Supplementary Fig. 12).

### Vigilance state analysis
ECoG or depth electrode LFP signals were analyzed in Spike2 software (Cambridge Electronic Design Limited, Cambridge, UK) or AccuSleep MATLAB-based interface[79], by segmenting the data into 5-second epochs, followed by visual inspection of the ECoG/LFP and EMG signals. Wakefulness was identified by desynchronized ECoG/LFP patterns and relatively high levels of EMG activity. Non-rapid-eye-movement (NREM) sleep was marked by high-amplitude, slow wave patterns accompanied by low EMG activity. Virtual channels derived from the ECoG/LFP signal, particularly delta power (0.5-4 Hz), were used to support NREM stage identification. However, in DS mice, as the increased delta activity during NREM is diminished, the EMG signal contributed to the classification. To distinguish rapid-eye-movement (REM) sleep from wakefulness, a theta (4−8 Hz) to delta power ratio of > 2.5, along with low EMG activity relative to NREM, was applied in the classification. Periods where wakefulness and NREM sleep alternated within 20 seconds were classified as transition states. Example recordings with hypnograms including all stages are shown Supplementary Fig. 3, whereas the hypnograms in Figs. 3−5 are shown without indication of these transitions for improved clarity. Transition periods, which did not allow for classification, were excluded from analysis. Two investigators analyzed the vigilance states independently, and only sections that were mutually agreed upon were considered for further analysis. For the data shown in Fig. 2, in WT mice, over 90% of the transitions from wake to NREM were agreed upon by the two investigators ($n = 34$ transitions from recordings of 6 WT mice). In DS mice, the two investigators agreed on ~ 70% of the transitions ($n = 37$ transitions from recordings of 8 DS mice). Artifacts, sleep arousals

(defined as increased short gamma activity accompanied by active EMG within sleep epochs), and epileptiform activities were carefully excluded to ensure accurate vigilance state scoring. Video recordings were consulted to verify sleep-wake classifications in cases of uncertainty.

## ECoG/LFP analyses of power spectral density

Power spectral density (PSD) was calculated using a fast Fourier transform with a Hann (cosine-bell) data window of 50% overlap (LabChart 8 software, AD Instruments, Colorado Springs, CO, USA). For each mouse, we calculated the PSD of the ECoG, or depth electrode LFP, during wakefulness and NREM, using the average value of the bilateral electrodes. The PSD of wakefulness was calculated from 30 s of recording, starting 2 min prior to the transition to NREM (in some cases, slight modifications of the timing were used to avoid movement artifact). The PSD of NREM sleep epochs was calculated from 30 s of recordings starting from one minute into NREM. Data from all transitions for each mouse were averaged prior to averaging across different mice, such that each mouse contributed one PSD value to the analysis. The ratio of delta was calculated as the sum of the power between 0.9 and 3.9 Hz, divided by the sum of the total power between 0.9 and 99 Hz.

## Immunohistochemistry, immunofluorescence and in situ hybridization

Immunohistochemistry, immunofluorescence, and in situ hybridization using RNAscope technology were performed as described previously[28]. Primary and secondary antibodies included chicken anti-GFP (1:1000, Abcam, ab13970, RRID:AB_300798), and Alexa Fluor® 488 donkey anti-chicken IgG (1:200, Jackson ImmunoResearch, Cat#703-545-155, RRID:AB_2340375). For colorimetric immunohistochemistry, sections were incubated with an avidin–biotin complex (Vector Laboratories, PK-6100, RRID:AB_2336819) and peroxidase activity was visualized using 0.05% 3,3'-diaminobenzidine (Sigma, D5637), and visualized using and visualized using Nanozoomer 2 Hamamatsu,Nikon Eclipse NI-E with a DS-Ri2 camera (4908 × 3264 px, 7.3 μm), and Leica Thunder microscope with a Leica K3C USB3 color camera (3072 × 2048, 2.4 μm pixels). Immunofluorescence images were acquired with a Zeiss LSM980 Airyscan microscope. RNAscope™ Multiplex Fluorescent V2 Assay (Advanced Cell Diagnostics, Cat#323100) was used to detect *Gad1* (Cat#400951) and *Scn1a* (Cat#434181).

## Statistical analyses

Data are reported as mean ± standard error (SE). All statistical tests were carried out via GraphPad Prism. The details of the specific test and the *p*-values are listed in the Figure legend, with the full description listed in Supplementary Data 1. Differences were considered significant at $p < 0.05$. The illustrations in Figs. 1a, 3c, i, 4a, e, f, 5a, Supplementary Fig. 6a and Supplementary Fig. 11a were created in BioRender. Rubinstein, M. (2026) https://BioRender.com/ol3g4vn

## Reporting summary

Further information on research design is available in the Nature Portfolio Reporting Summary linked to this article.

## Data availability

Source data are provided in this paper.

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

## Acknowledgements

This work is dedicated to the memory of Dr. Saja Fadila, whose dedication, talent, and scientific insight were central to this study. We acknowledge the financial support of The Israel Science Foundation (#214/22, M.R.), Recanati Foundation, Gray Faculty of Medical & Health Sciences (M.R.); Marguerite Stolz Research Fellowship, The Gray Faculty of Medical & Health Sciences (M.R.); The American Dravet Syndrome Foundation (M.R., E.A.T., E.J.K.); The Dutch National Epilepsy Foundation 22-07 (G.K., E.A.T.); The Yoran Institute for Human Genome Research at Tel Aviv University (supported the Ph.D. scholarship of S.F.). The imaging facility MRI at Montpellier, a member of the national infrastructure France-BioImaging infrastructure supported by the French National Research Agency (ANR-10-INBS-04, Investments for the future). The Réseau d'Histologie Expérimentale de Montpellier - RHEM facility is supported by SIRIC Montpellier Cancer (Grant INCa_Inserm_D-GOS_12553), the European Regional Development Fund, and the Occitanian region (FEDER-FSE 2014-2020 Languedoc Roussillon). We thank the members of the Rubinstein lab for constructive comments during the course of the study.

## Author contributions

Conceptualization: S.F. and M.R.; Methodology: S.F., G.K., H.M., A.M., S.R., M.B., B.B., I.G.D.R., E.J.K., E.A.T., and M.R.; Software: G.K. and S.R.; Validation: S.F., G.K., E.A.T., and M.R.; Formal Analysis: S.F., G.K., S.R., E.A.T., and M.R.; Investigation: S.F., G.K., H.M., A.M., S.R., M.B., and I.G.D.R.; Resources: B.B., I.G.D.R., and E.J.K.; Writing – original draft: S.F. and M.R.; Writing – Review & Editing: S.F., G.K., E.J.K., E.A.T., and M.R.; Supervision: E.A.T. and M.R. Funding Acquisition: E.A.T. and M.R.

## Competing interests

The authors declare no competing interests. S.F., B.B., I.G.D.R., E.J.K., and M.R. are co-inventors on a patent application covering the use of CAV2 vectors for the delivery of an SCN1A expression cassette to provide exogenous Na$_V$1.1 activity for the treatment of Dravet syndrome and related disorders. The patent applicant is the French National Center for Scientific Research (CNRS), Université de Montpellier, together with Ramot at Tel Aviv University Ltd., and Fundacion Para Investigacion Medica Aplicada. The patent application was filed as GB2205299.7 and published as GB2621102A, with international extension PCT/IB2023/053703), and is currently pending. In the present manuscript, CAV2-SCN1A was used as an experimental tool to assess whether exogenous SCN1A expression can correct sleep and temperature deficits in DS mice.
