## [Transparent Peer Review file · Nature Communications]

Disrupted temperature-sleep coupling mechanism in a Dravet syndrome mouse model

Corresponding Author: Professor Moran Rubinstein

Version 0:

Reviewer comments:

Reviewer #1

(Remarks to the Author)

I liked this manuscript, and I think the results are interesting and give important insights into aspects of the pathology of Dravet syndrome. The basic idea that the authors discover is that in mice that have only one functioning SCN1A allele (a Dravet model) the phenomenon of nest building, warmth-induced NREM sleep and the concurrent hypothermia as the mice enter NREM is missing; this sleep-temperature phenotype can be rescued by expressing the SCN1A channel back into the preoptic hypothalamus. In general, the manuscript is well written, and the experimental logic and description are easy to follow, and the Intro and Discussion give balanced viewpoints. Nevertheless, I do have some specific suggestions/comments:

1. line 89, line 105. Heated pad set to thirty degrees. Does this actually make the air temperature in the beaker thirty degrees, or is the air just warmer but not at 30 degrees C? I guess the key thing is that the skin of the mice is being warmed rather than that the air temperature being at 30 degrees? And later experiments, line 358, are done at 36 degrees on the warming pad, considerably hotter. Why are the different temperatures used? Please clarify.
 2. Does SCN1A re-expression in the hypothalamus of SCN1A mice rescue the nesting phenotype? Do the mice now build nests? Maybe not, but it would be interesting to know.
 3. Does SCN1A re-expression in the hypothalamus of SCN1A mice rescue the “temperature decrease with sleep” phenotype that is shown in Fig. 2F and L? Again, if not, fine, but it would be good to know.
 4. I found the “sleepiness index” quite confusing. A number of parameters are condensed into a “sleepiness scale”, but I was not convinced how useful this was. It was unclear how all the factors blend in, and what gets under or overestimated. Maybe a rethink is needed here, or simply take that part out.
I don't really understand the graphs in Fig. 3F, and Fig. 4M. They left me confused. In Fig. 3F what is the x axis? It is not labelled. The simple graphs and parameters in e.g. Fig. C and J and Fig. 4J are straightforward in contrast.
 5. In Fig. 4, I think expressing SCN1A in wild-type hypothalamus and comparing it with SCN1A re-expression in Dravet mice is the wrong comparison. Overexpressing a sodium channel might give a phenotype in itself. I would suggest, instead, that Fig. 4A to G (the WT results) could be a Supp. Figure. On the other hand, parts H through to N of Fig 4 (expression of SCN1A in Dravet hypothalamus) should be moved into Fig 3, so that we have wild-type (top part), Dravet phenotype middle part, and rescue of phenotype with SCN1A in hypothalamus in bottom panel. This seems to be the correct comparison. So, I suggest redesigning Fig. 3 and getting rid of the current Fig. 4
 6. CNO expts.
 - a. 10 mg/kg CNO is at the top-end and I would think too high to be sure the effect is not due to conversion to clozapine, which induces sedation. What does 10 mg/kg CNO do to a control mouse? Important to check it does not induce, to some extent, cause sleepiness.
 - b. Fig. 5B. More detail is needed for the anatomical expression of the virus. It seems surprising that expression is just in a micropatch of LPO? I would think that might be possible in a rat, but not a mouse hypothalamus. The photomicrograph just looks odd to me. More extensive anatomical characterization should be given of viral expression. In the schematic in part A, please indicate the timing of when CNO or saline is given prior to the mice being placed in a warm environment.
 - c. Given that the hM3Dq receptor is expressed pan-neuronally in the preoptic hypothalamus, it seems surprising that the same result is always produced. I would think that sometimes there would be more expression in wake-promoting preoptic neurons, depending on each animal where the virus is transduced exactly, and wakefulness is produced with CNO. There must be quite a lot of variability here? (glutamate neurons in the preoptic can produce wake when excited).
- Signed: William Wisden

(Remarks on code availability)

Reviewer #2

(Remarks to the Author)

In this novel study, Dr. Rubinstein and colleagues shed light on the understudied yet crucial problems of thermal dysregulation and sleep disturbances in Dravet syndrome. Notably, they establish a connection between these two deficits in a mouse model of the disorder. Their findings reveal that Dravet mice have a lower baseline body temperature. Furthermore, through integrated EcoG, LFP, and temperature monitoring, they discovered that, unlike wild-type mice, which experience a temperature drop transitioning from wakefulness to NREM sleep, Dravet mice do not. Intriguingly, increasing body temperature induced sleep in wild-type mice but failed to do so in Dravet mice. Interestingly, when they restored Nav1.1 expression using a CAV2-Nav1.1 vector that they previously developed or induced neuronal activation using chemo-genetics in the anterior hypothalamus, warmth-induced sleep was rescued in the Dravet mice.

This work is novel, well-designed, and well-written and contributes to the field by highlighting the contribution of previously unexplored brain areas to the multifaceted phenotypes of Dravet syndrome. Publication of this important paper will be very valuable to the field. That said, there are some issues that need to be integrated or better characterized to maximize the informative power and the impact of the work.

1. The authors focused on the transition from wakefulness to NREM sleep to investigate the link between temperature dysregulation and sleep disturbances. However, to have a complete picture, it would be worth to study additionally the link between core body temperature alterations and the transition from NREM to REM phase. This is likely something not so difficult to do as they have already performed the recordings.
2. The results of the two rescue experiments with injection of CAVs expressing either Nav1.1 or hM3D are impressive. However, they could be more informative if the authors could better characterize the percentage and extent of neuronal transduction in the hypothalamus and interconnected brain areas, considering the retrograde properties of CAVs.
3. In which neuronal cell types is Nav1.1 expressed in the hypothalamus? RNAscope or analysis of available single cell data set of adult mouse brain could help.

(Remarks on code availability)

I don't have the experience to test the code.

Reviewer #3

(Remarks to the Author)

In this report, Fadila et al., seek to establish a connection between sleep homeostasis and thermoregulation in an SCN1A haplo-insufficient mouse, carrying a Dravet Syndrome-associated mutation. Mouse models of DS recapitulate several of features of disease including seizures, impaired sleep and poor thermoregulation. Changes in core body temperature coincide with sleep/wake state transitions and increasing ambient temperature can promote NREM sleep. The authors hypothesize that sleep and temperature homeostasis disturbances are linked in DS mice and seek to understand the mechanisms by which they are coordinated. There is potentially dual value in this approach as it stands to be informative regarding the relationship between sleep and thermoregulation in the disease but also could reveal principles mechanistically linking SCN1A function to sleep and temperature homeostasis. While this is an interesting manuscript with some very nice preliminary findings, there are many methodological concerns (detailed below) including a lack of appropriate controls, inappropriate statistical analyses, and a lack of experimental detail as well as very limited mechanistic insight rendering the manuscript of limited impact in its current form.

Our specific concerns and suggestions are laid out below more or less organized by Figures.

Figure 1

// The authors claim that "DS mice increased their core more rapidly than WT mice" (line 90), however no quantification of the rate of change in core body temperature is provided. When mice are returned to RT to recover, the genotypes have different core body temperatures. However, it is curious that the authors do not highlight that both genotypes 'overshoot' their temperature recovery, ending up at a temperature below the baseline core body temperature shown in 1B and 1C. Do both WT and DS mice have impaired recovery in this model? .

The authors next evaluate nesting behavior in WT vs DS mice because nests can stabilize core body temperature during sleep. DS mice may not build nests as complex as their WT littermates however, the significance of these findings at this juncture (or anywhere for that matter in the manuscript) is unclear. While pre-sleep behaviors in mice are now well recognized. In the context of the current manuscript, a defect in these behaviors raises the possibility of important confounds: for example, perhaps DS mice defect in nesting contributes to a lack of movement that would raise body temperature prior to sleep. This possibility is not addressed.

//Moreover, the localization of these pre-sleep behaviors is still not well circumscribed in the brain.

// It is unclear how these findings fit with previous and subsequent data in the manuscript. How does nesting behavior related to core body temperature in DS mice, sleep and hypothalamic activity? If nesting behavior is abnormal in DS mice, could this argue that SCN1A mutations disrupt circuits involved in behaviors which regulate temperature? This line of questioning is distinct from hypothesis that SCN1A acts in key brain regions directly involved in regulating core body temperature (i.e. hypothalamus).

Figure 2

The authors next evaluate sleep and the change of core body temperature during wake -> NREM transitions in WT and DS mice. They find that while WT mice have a higher delta power in NREM compared to wake, DS mice do not exhibit any change in delta power. In addition, while WT mice experience a decrease in core body temperature at wake -> NREM transitions, there is no change in DS mouse core body temperature at transition states.

//There are major technical concerns with the sleep recordings. It is unclear from the methods section at what time of day the recordings were done and how the mice were housed during the recording session.

// Were animals protected from environmental disturbances? Were they acclimated to the recording chambers or temperature probes before recording sleep?

// Additionally, the recovery period post-surgery/implantation seems relatively short (2-3 days).

// The statistical analysis of the PSD is not appropriate. It is advised to conduct a 2-way ANOVA, comparing WT-WAKE, WT-NREM, DS-WAKE and DS-NREM. From Figure S1, it appears that the authors instead conducted t-tests to compare within genotypes only. The authors state "In DS mice, the increase in delta power was smaller than in WT animals" (line 160). With the current statistical analysis, they cannot make such a claim (unless we missed this somewhere).

// As a general comment for all sleep recordings done in the report, there are often poor representative ECoG & EMG recording traces. Figure 2A, for example, displays ECoG and EMG traces in 'wake' and 'NREM' which are entirely indistinguishable from each other.

Figure 3

// the authors argue that WT mice exhibit an increase in delta power after heating, but it should be addressed whether this effect is driven by one mouse which seems to have a significant change (3D).

// The authors also employ a 'sleepiness scale' to evaluate the relative sleepiness of WT and DS mice after heating.

However, it is unclear as to why the development or use of this 'sleepiness scale' is required. We question whether the use of EMG signal as a marker for sleepiness, in itself, is arbitrary and inappropriate. For example, a mouse with a severed spinal cord or with absence of spinal motor neurons may have different/absent EMG signals but no change in "sleepiness" but this scale would render a "sleepier" mouse simply as a function of immobility. Without validation or generalizability of such a scale, the data from it is hard to understand. Furthermore, this scale not used in Figure 2 recordings -- why not? For this and subsequent figures (3, 4 and 5) the authors also no longer separate a PSD analysis from wake vs NREM and it is unclear why (in line 227). It seems that the main effect of PSD delta power was in NREM stages (figure 2) and they may be missing an opportunity to have stronger effects.

Figure 4

The authors test whether expression of SCN1A in the anterior hypothalamus can restore warmth-mediated sleep in DS mice. They argue that CAV-mediated SCN1A expression in DS mice restores the increased NREM time and delta power induced by heating.

// However, this experiment is lacking appropriate control (empty virus not "untreated") and statistical analyses. While it appears the authors had an 'untreated' group (showed in supplements) they do not include these groups in their statistical analyses.

// And again, the authors fail to statistically compare WT and DS mice against each other. They could do this by statistically comparing the change in delta ratio from RT -> 36°C in each injection x genotype group.

Figure 5

The authors test whether chemogenetic activation of the anterior hypothalamus using activating DREADDS can rescue warmth-mediated sleep in DS mice.

//The rationale for the experiment is only partially clear. What is known about the activity of the neurons in anterior hypothalamus in DS mice? Is the activity in the haplo-insufficient model lower or higher than WT mice? In which cells? Testing this would support the DREADD experiments.

// There is again a lack of proper control (empty vector) and appropriate approach to statistical analysis. Including a CNO only control would be ideal as the effects observed could be CNO or hypothalamus-mediated.

// there is no confirmation that CNO administration actually had an effect on anterior hypothalamus activity.

Other major concerns

- Lack of description about the haplo-insufficient DS mouse model
- Varied ages for the mouse studies
- baseline temperature difference is gone in older animals
- many seizures in the younger animals, do any of them die?
- Lack of investigation and discussion about potential metabolic contributions to differences in basal and sleep-state transition associated thermoregulation
- No investigation into the activity of the hypothalamus in DS mice, this could link hypothalamic involvement to the temperature-sleep transitions
- There is also occasional inconsistency in the PSD findings – sometimes the main significant effect of the experimental variable is seen in delta power and sometimes it is in delta ratio, but there is a lack of discussion about this.

(Remarks on code availability)

Reviewer #4

(Remarks to the Author)

I co-reviewed this manuscript with one of the reviewers who provided the listed reports. This is part of the Nature

Communications initiative to facilitate training in peer review and to provide appropriate recognition for Early Career Researchers who co-review manuscripts.

(Remarks on code availability)

Version 1:

Reviewer comments:

Reviewer #1

(Remarks to the Author)

The authors have made very good efforts to address all my points. I think the manuscript and presentation are improved and the work is an interesting and important contribution.

Signed: William Wisden

Reviewer #2

(Remarks to the Author)

The authors addressed all my concerns.

Reviewer #3

(Remarks to the Author)

The authors have made a series of commendable revisions that have satisfied my concerns with the initial manuscript. Congratulations.

Response to Reviewers' comments on Fadila et al. NCOMMS-25-18755.

We thank the Reviewers for their insightful and helpful comments and provide point-by-point responses below in blue font, with references to changes in the main text indicated in grey font. In the revised manuscript, textual changes based on the comments are indicated with tracked changes. The reference to line numbering below refers to the revised manuscript without tracked changes active.

Reviewer #1 (Remarks to the Author):

I liked this manuscript, and I think the results are interesting and give important insights into aspects of the pathology of Dravet syndrome. The basic idea that the authors discover is that in mice that have only one functioning SCN1A allele (a Dravet model) the phenomenon of nest building, warmth-induced NREM sleep and the concurrent hypothermia as the mice enter NREM is missing; this sleep-temperature phenotype can be rescued by expressing the SCN1A channel back into the preoptic hypothalamus. In general, the manuscript is well written, and the experimental logic and description are easy to follow, and the Intro and Discussion give balanced viewpoints. Nevertheless, I do have some specific suggestions/comments:

Answer: Thank you for your helpful comments and suggestions, which we carefully addressed in the revised manuscript.

1. line 89, line 105. Heated pad set to thirty degrees. Does this actually make the air temperature in the beaker thirty degrees, or is the air just warmer but not at 30 degrees C? I guess the key thing is that the skin of the mice is being warmed rather than that the air is just warmer but not at 30 degrees C? And later experiments, line 358, are done at 36 degrees on the warming pad, considerably hotter.

Why are the different temperatures used? Please clarify.

Answer: When the heated pad was set to 30°C, the temperature at the bottom of the beaker where the mouse was placed reached approximately 26°C. When the pad was set to 36°C, the temperature at the bottom of the beaker reached ~29°C.

The rationale for using two different temperature settings relates to the age-dependent sensitivity of DS mice to heat-induced ('febrile') seizures, which varies between the third and fifth weeks of life. In the experiment shown in Fig. 1, we examined DS mice at their 4th week of life (P21-P25), when their susceptibility to febrile seizures is highest. For this reason, a modest increase in temperature (i.e., with the heating pad set at 30°C, yielding an ambient temperature of 26°C at the bottom of the beaker) was chosen. Even under these moderate warming conditions, approximately 25% of the mice experienced heat-induced seizures. By contrast, for the experiments shown in Figs. 3-5, the mice were older, five to six weeks old, at which point their sensitivity to febrile seizures is greatly reduced. Therefore, we chose to use a higher heating pad temperature (i.e., 36°C, yielding ~29°C at the bottom of the beaker) to provide a more robust warmth stimulus that more closely mimics conditions used in prior studies investigating the sleep-promoting effects of elevated ambient temperature.

This information has now been included in the respective Methods section (page 18, lines 5-12)

To test the effect of temperature on sleep promotion, the mice were recorded in two blocks: the first one at room temperature for one hour, after which the recording chamber was moved and placed over a heated pad set to 36°C (TCAT-2DF, Physitemp Instruments, Clifton, NJ USA), and a second block was recorded for another hour. Under these conditions, the air at the bottom of the beaker was at 29°C. Note that for these experiments, since the DS mice were older than five weeks, a higher ambient temperature of 36°C was used (and not the 30°C used for the younger DS mice in Fig. 1), given their reduced sensitivity to thermally induced seizures at this age.

and Results sections (page 7, lines 16-32).

A warm ambient temperature (32-36°C) promotes sleep through a mechanism that involves the activation of heat-sensing neurons in the hypothalamus (Harding et al., 2019; Kroeger et al., 2018). Given the deficits in temperature regulation upon the rise to a moderate ambient temperature of 30°C, and the sleep transition-related changes in delta power in DS mice, we set out to investigate whether warmth-induced somnogenesis is impaired in DS mice. Depth LFP and EMG activity were recorded in 5-6-week-old WT and DS mice. Recordings were first obtained for one hour at room temperature, followed by an additional hour under elevated ambient temperature achieved by placing the cage on a heating pad set to 36°C, resulting in a temperature of 29°C at the bottom of the recording beaker (Fig. 3A-C). Because DS mice at this age show reduced susceptibility to febrile seizures (Almog et al., 2021; Mavashov et al., 2023), we reasoned that a somewhat stronger warming stimulus could be applied compared to the 30°C condition used in younger animals (Fig. 1). However, to minimize seizure risk, we avoided the higher temperatures (32-36°C) used in previous studies (Harding et al., 2019; Kroeger et al., 2018) making 29°C a deliberate compromise between physiological relevance and safety in this model.

2. Does SCN1A re-expression in the hypothalamus of SCN1A mice rescue the nesting phenotype? Do the mice now build nests? Maybe not, but it would be interesting to know.

Answer: We did not assess nesting behavior in the original dataset of hypothalamic CAV2-SCN1A-injected mice. To address this question, we performed a new set of experiments, in which the effects of hypothalamic injection of CAV2-SCN1A were compared with those of the control vector CAV2-GFP in both WT and DS mice. Seven to ten days after vector injection, we tested the nest-building abilities of the mice. As shown in the newly added Supplementary Fig. 10, and mentioned in the Results section on page 14, lines 1-12, CAV2-SCN1A injection into the hypothalamus failed to correct this DS-associated phenotype. (pages 8-9, lines 28-7)

To directly control for the effects of the hypothalamic CAV2 injections, a separate cohort of WT and DS mice was injected with a control vector containing a reporter gene CAV2-GFP, or with CAV2-SCN1A.

...

In the cohort of CAV2-GFP or CAV2-SCN1A injected mice, we also evaluated nest-building behavior. Seven to ten days after vector injection, CAV2-SCN1A did not rescue the impaired nest-building abilities of DS mice (Supplementary Fig. 10).

Supplementary Fig. 10

CAV2-SCN1A injection into the hypothalamus does not correct nest-building abilities in DS mice.

Examples of the nesting material after a night in the home cage (left) and nest-building scores (right) showing reduced nest-building behavior in DS mice injected with CAV2-GFP or CAV2-SCN1A (WT: CAV2-GFP, n = 7. WT: CAV2-SCN1A, n = 8. DS: CAV2-GFP, n = 6. DS: CAV2-SCN1A, n = 6. $p < 0.0001$, 2-way ANOVA. The results of Holm-Sidak post hoc analyses are depicted on the graph.

3. Does SCN1A re-expression in the hypothalamus of SCN1A mice rescue the “temperature decrease with sleep” phenotype that is shown in Fig. 2F and L? Again, if not, fine, but it would be good to know.

Answer: Unfortunately, in the original experiments of hypothalamic CAV2-SCN1A-injected mice, shown in Fig. 3, we did not include body temperature monitoring. We now included temperature recordings in the additional experiments, referred to also in our answer to the previous question, that were performed in mice injected in the hypothalamus with CAV2-SCN1A or CAV2-GFP.

For these experiments, WT and DS mice were injected in the hypothalamus with the control vector, CAV2-GFP, at the age of P22-P26, similar to the experiments shown in the original Fig. 3. After assessing their nest-building abilities (as described above), the mice underwent another surgery to implant the ECoG electrodes. To address whether the “temperature decrease with sleep” is restored by hypothalamic SCN1A expression, we assessed the effect of hypothalamic CAV2-SCN1A on sleep properties and temperature at RT using 4-h recordings at RT in 5-6 week-old mice. These recordings, similar to the original experiments in naïve mice shown in Fig. 2, were performed 2-3 days after ECoG electrode implantation.

First, we assessed the contribution of delta frequencies to the overall ECoG spectral power during wakefulness (2 minutes prior to the transition to sleep) and during the first minute of NREM sleep, and examined changes in core body temperature across these transitions. In this set of experiments, however, the characteristic temperature decrease at sleep onset was not evident in any of the experimental groups, including WT mice injected with either CAV2-GFP or CAV2-SCN1A. Since similar effects were observed across all experimental groups, regardless of genotype or vector type (CAV2-GFP or CAV2-SCN1A), the discrepancy between these findings and those of the untreated WT and DS mice presented in Fig. 2 appears attributable to differences in experimental procedures. The mice in Fig. 2 underwent a single surgical intervention (either ECoG implantation at 4 weeks of age, or LFP implantation at 5 weeks), whereas the mice of the additional set of experiments injected with CAV2-GFP or CAV2-SCN1A, underwent two surgeries, i.e., one for vector injection at 4 weeks of age and

a second one for ECoG implantation at 5 weeks of age. Furthermore, in the experiments shown in Fig. 2, invasive LFP electrode implantation was performed in 5-week-old mice, whereas viral injections targeting the same region were carried out in younger mice (4 weeks old). Thus, although both procedures involved mechanical disturbance of the hypothalamus, it is possible that younger mice are more vulnerable to such perturbations, leading to disruptions in temperature drop at NREM onset, independent of SCN1A expression.

A description of these findings has been added to the Results section (pages 8-9 , lines 28-24), and is shown in new figures Fig. 4a-e, Supplementary Fig. 11.

To directly control for the effects of the hypothalamic CAV2 injections, a separate cohort of WT and DS mice was injected with a control vector containing a reporter gene CAV2-GFP, or with CAV2-SCN1A

....

In the cohort of CAV2-GFP or CAV2-SCN1A injected mice, we also evaluated nest-building behavior. Seven to ten days after vector injection, CAV2-SCN1A did not rescue the impaired nest-building abilities of DS mice (Supplementary Fig. 10). Next, ECoG electrodes were implanted, and the mice were allowed to recover for 2-3 days before we proceeded with ~4 h ECoG and temperature recordings at RT. Also in this cohort of mice, CAV2-SCN1A expression in the hypothalamus restored the increase in delta ratio during the transition to NREM sleep, which was absent in DS mice injected with the control vector (CAV2-GFP, Fig. 4a-e). Of note, these analyses were conducted using cortical ECoG rather than the hypothalamic LFP recordings used for the CAV2-SCN1A experiments shown in Fig. 3, confirming that the correction of sleep-related oscillatory activity by CAV2-SCN1A is also reflected at the cortical level. Core temperature measurements were included in this new cohort of mice to test whether hypothalamic SCN1A expression may also correct the drop in body temperature at NREM onset. However, in contrast to the data from naïve untreated mice shown in Fig. 2, no significant temperature decrease was detected after NREM onset in either WT or DS mice injected hypothalamically with either CAV2-GFP or CAV2-SCN1A. The discrepancy between these results and those presented in Fig. 2 may be attributable to the different experimental procedures, as the mice depicted in Fig. 2 underwent only one surgical process, for electrode implantation (ECoG at the age of 4 weeks, or LFP at the age of 5 weeks), while the mice shown in Fig. 4 underwent two surgeries, one for vector injection at 4 weeks and a second one for ECoG implantation at 5 weeks.

4. I found the “sleepiness index” quite confusing. A number of parameters are condensed into a “sleepiness scale”, but I was not convinced how useful this was. It was unclear how all the factors blend in, and what gets under- or overestimated. Maybe a rethink is needed here, or simply take that part out.

Answer: Thank you for this constructive feedback. Following your advice, we removed this index and instead quantified the percentage of NREM sleep at RT and a warm environment (i.e., when the mice are placed on a heating pad set to 36°C). These data are now shown in Figs. 3-5. In addition, following the advice of Reviewer #3 (and to align with our measurements shown in Fig. 2), we have added in Figs. 3-5: a quantification of the delta ratio during wakefulness and NREM. To align with the results of Fig. 2, the analyses of wakefulness were conducted for the period 2 minutes prior to the transition to NREM, and the NREM analyses used the 1-minute period into NREM.

I don't really understand the graphs in Fig. 3F, and Fig. 4M. They left me confused. In Fig. 3F what is the x axis? It is not labelled. The simple graphs and parameters in e.g. Fig. C and J and Fig. 4J are straightforward in contrast.

Answer: These graphs depicted the sleepiness index in individual mice (so the x-axis represented different mice). As noted above, these 'sleepiness index' measurements have now been replaced with measurements of the percentage of NREM and the delta ratio at wake and NREM.

5. In Fig. 4, I think expressing SCN1A in wild-type hypothalamus and comparing it with SCN1A re-expression in Dravet mice is the wrong comparison. Overexpressing a sodium channel might give a phenotype in itself. I would suggest, instead, that Fig. 4A to G (the WT results) could be a Supp. Figure. On the other hand, parts H through to N of Fig 4 (expression of SCN1A in Dravet hypothalamus) should be moved into Fig 3, so that we have wild-type (top part), Dravet phenotype middle part, and rescue of phenotype with SCN1A in hypothalamus in bottom panel. This seems to be the correct comparison. So, I suggest redesigning Fig. 3 and getting rid of the current Fig. 4

Answer: Thank you for this suggestion. As advised, we are now showing untreated and CAV2-SCN1A-injected DS mice in the same Figure (i.e., Fig. 3). Moreover, we added a direct statistical comparison between untreated WT, untreated DS, and CAV2-SCN1A-injected DS mice, which is shown in Supplementary Fig. 6b. As shown in this figure, there was no statistical difference between naïve and CAV2-SCN1A-injected WT mice.

Supplementary Fig. 6

A full statistical comparison between different experimental groups shown in Figs. 3, 4.

(a). The experimental paradigm, used for examining the effect of warmth on sleep promotion.

(b-c). The same data shown in Figs. 3d and 3j are replotted here to highlight genotype differences. (2-way ANOVA. The results of Holm-Sidak post hoc analyses are depicted on the graph).

(d). The data shown in Figs. 4f are shown are depicted again here with the inclusion of the full statistical analysis to highlight also genotype differences. (2-way ANOVA. The results of Holm-Sidak post hoc analyses are depicted on the graph).

* $p < 0.05$; ** $p < 0.01$; *** $p < 0.001$

6. CNO expts.

a. 10 mg/kg CNO is at the top-end and I would think too high to be sure the effect is not due to conversion to clozapine, which induces sedation. What does 10 mg/kg CNO do to a control mouse? Important to check it does not induce, to some extent, cause sleepiness.

Answer: We selected this dosage based on a previous paper that tested the effect of chemogenetic on seizure threshold (Mattis et al., 2022; PMID: 35212623). To directly test the possibility that this dosage may cause sleepiness, we examined the effect of 10 mg/kg CNO on locomotor activity in four untreated awake WT mice assessed for a period of 10 minutes, starting 15 minutes after CNO injection. This time frame aligns with the exclusion from analysis obtained from the first 15 minutes after CNO injection (Jendryka et al., 2019, PMID:30872749; Vardy et al., 2015, PMID:25937170) that we adhered to for the warmth-induced somnogenesis experiments, information which we have now included in the Methods section (page 18, lines 17-20).

Locomotor activity of WT mice, without DREADD expression, was not affected by 10 mg/kg CNO. Specifically, 15 min after CNO injection, locomotor activity of the mice was monitored for 10 minutes in an open field of 50 by 50 cm (Supplementary Fig. 12).

And the Results section (page 18, lines 1-3):

This dosage of CNO did not alter locomotor activity, as assessed in an open field test in WT mice without DREADD expression (Supplementary Fig. 12), indicating the absence of a nonspecific sedation effect at this dose³⁷.

Supplementary Fig. 12

CNO administered at 10 mg/kg had no effect on the locomotor activity of WT mice. WT mice were placed in a 50 x 50 cm open field arena 15 min after CNO injection, and their locomotor activity within the arena was recorded for 10 min. Four days later, the same mice were injected with saline, and their locomotor activity was recorded again 15 min after saline injection over 10 min. The similar distance covered under both conditions indicates that CNO at this dose does not have a non-specific sedative effect.

And the Discussion section (page 14, lines 21-25) :

A limitation of the present study is that we did not directly assess how the effects of the DS *Scn1a* mutation and hypothalamic CAV2-mediated SCN1A expression, or alternatively chemogenetic stimulation, influence neuronal firing and excitability in the anterior hypothalamic region. Future studies addressing these questions could provide deeper mechanistic insight into the neuronal basis of sleep and thermoregulation deficits in DS.

Moreover, while DS mice generally showed an enhanced delta contribution in NREM after CNO administration, we detected variability in the magnitude and the direction of the response to CNO, with one mouse displaying a reduction in NREM delta power at 36°C (Fig. 5J green), suggesting that the effect differ, as proposed by the Reviewer (see below) based on the specific neuronal population which was transduced, rather than a non specific effect.

b. Fig. 5B. More detail is needed for the anatomical expression of the virus. It seems surprising that expression is just in a micropatch of LPO? I would think that might be possible in a rat, but not a mouse hypothalamus. The photomicrograph just looks odd to me. More extensive anatomical characterization should be given of expression.

Answer: Unfortunately, we did not obtain better expression images with CAV2-hM3D-IRES miCitrine in the brain. The amount of mCitrine, which is the ORF after the IRES, is very low (Mizuguchi et al. 2000; PMID:10933956). And, like numerous labs, we also attempted to use commercial antibodies against hM3D, but in our hands, the signal was weak and not specific. We are also unaware of any study showing convincing hM3D immunoreactivity in vivo.

Using CAV2-hM3D-IRES- mCitrine in E1-transcomplementing cells (DK-E1), we observe mCitrine expression (added as Supplementary Fig. 9i-n). Detection of mCitrine in DK-E1 cells seeded as a monolayer, with transduction of the majority of cells on the coverslip, is easier than detecting transduced neurons in a brain following in vivo expression. These data indicate that the vector quality (physical to infectious particle ratio) is excellent and that transgenes from CAV2-hM3D-IRES-mCitrine are expressed. Briefly, DK-E1 cells were incubated with 50 particles of CAV2-hM3D IRES-miCitrine/cell. 24 h post-incubation, mCitrine and hM3D immunofluorescence were examined.

Supplementary Fig. 9

(i-n). CAV2-hM3D IRES miCitrine expression in the DK-E1 cells. DK-E1 cells were seeded as a monolayer and incubated with CAV2-hM3D-IRES-mCitrine (50 particles/cell) for 24 h, followed by immunofluorescence. mCitrine immunofluorescence (green), and hM3D (red).

(i) mCitrine; (j) hM3D; (k) mCitrine and hM4D; (l) DAPI and mCitrine; (m) DAPI and hM3D, (n) DAPI, mCitrine and hM3D). Anti-GFP (1:1000, Abcam, ab13970, RRID:AB_300798) was used to detect mCitrine. HPA024106 (1:100, Sigma) was used to detect hM3D.

Scale bar: 10 μ M

In addition, we injected three WT mice, in the hypothalamus, with a reporter vector, CAV2-CMV-mCitrine, which harbors the CMV promoter driving mCitrine. CAV2-CMV-mCitrine injections led to robust mCitrine immunoreactivity in the hypothalamus and connecting regions.

These data were not added to the manuscript to avoid confusion, as the expression levels of mCitrine using CAV2-CMV-mCitrine reporter vector are much higher than those of CAV2-hM3D-IRES mCitrine (which enabled the easy detection of the vector in the brain). Yet it indicates the expected biodistribution. See the full distribution of the biodistribution for this vector in our response to Reviewer #2, point 2; the panel depicted here corresponds to D).

mCitrine expression in the hypothalamus of a WT mouse following injection of CAV2-CMV-mCitrine to the anterior hypothalamus

Additionally, in response to the other Reviewer's comments, we have added a more detailed histological analysis using CAV2-NSE-mCitrine (mCitrine is driven by the NSE promoter, which also drives the expression of SCN1A in CAV2-SCN1A). Both promoters drive transgene expression in excitatory and inhibitory neurons. These data using CAV2-NSE-mCitrine are presented in Fig. 4B,C, Fig. S8, and Fig. S9, and corresponding statements have been added to the Results (pages 8-9, lines 30-3)

The biodistribution was examined in mice following CAV2-GFP injection (WT and DS, n = 3 for each genotype), which were also used for behavioral and electrophysiological examinations, as well as in a separate group of three WT mice injected with the control vector CAV2-mCitrine, which were only used for immunohistochemistry. The biodistribution analysis demonstrated expression in excitatory and inhibitory neurons in the hypothalamus and, due to CAV2's retrograde transport properties, also in interconnected brain regions (Fig. 4a, b, Supplementary Figs. 8 and 9).

With the technical difficulties to demonstrate CAV2-hM3D-IRES mCitrine, we would like to share additional evidence to support the functional expression of this vector in the brain

This evidence comes from a separate project in which the same DREADD vector, CAV2-hM3D-IRES-mCitrine, was injected into the thalamus and hippocampus of DS mice. The goal in that project is to compare the effects of acute DREADD activation to those of CAV2-SCN1A in these regions. CAV2-SCN1A injections into the thalamus and hippocampus were previously shown to correct both epileptic and cognitive deficits in DS mice (Fadila et al., 2023; PMID: 37192002). The experimental procedures in this other project differ with respect to the targeted brain regions thalamus & hippocampus, vs. hypothalamus, recording configuration (i.e., ECoG in this other project vs. hypothalamic LFP for the DREADD experiments in the

current manuscript) and the DREADD agonist that was used (DCZ in this other project vs. CNO in the current manuscript).

In addition, the overall outcome in this other project concerned an aggravation of DS phenotypes following DREADD activation, instead of an improvement as observed for the correction of warmth-induced somnogenesis in the present manuscript. Despite the differences, however, these ongoing studies demonstrate that the same DREADD vector can elicit DS-specific effects based on behavioral and ECoG spectral power features, supporting the functional expression of this vector in the brain, as shown in the Figure below. Specifically, in DS mice, chemogenic stimulation resulted in increased anxiety, shown as reduced time at the center of the open field arena, as well as a reduction in the contribution of delta frequencies to the overall ECoG spectrum, along with an increase in the contribution of beta frequencies. The observation that no change in open-field activity or ECoG activity was observed in WT mice following chemogenic stimulation, while these changes were observed in DS mice, indicates a specific effect of DREADD activation in hippocampal and thalamic regions, supporting the functional effect of CAV2-hM3D-IRES-mCitrine in vivo.

Data from a parallel project involving DREADD injection into the thalamus and hippocampus. A, B) Following saline or DCZ (0.1 mg/kg), the locomotor activity (A, WT: n = 9; DS: n = 7 p = 0.4, 2-way ANOVA), and the location of the mice within the center of the arena (B) were examined (p = 0.002, 2-way ANOVA. The results of Holm-Sidak post hoc analyses are depicted on the graph)

C,D) The power of ECoG in WT (C, WT: n = 5; p = 0.7, 2-way ANOVA) and DS mice (D, DS: n = 3, p = 0.03, 2-way ANOVA) The results of Holm-Sidak post hoc analyses are depicted on the graph), at the indicated frequency bands, following DREADD activation.

In the schematic in part A, please indicate the timing of when CNO or saline is given prior to the mice being placed in a warm environment.

Answer: We have added to the legend of Fig. 5 that CNO or saline was administered immediately (t=0 min) prior to placing the mouse in the warm environment, as was stated already in the Methods section, and we added this information also to the legend of Fig.5.

Fig. 5 legend

(a) A schematic representation of the experimental design. Saline injection was given right before the mouse was placed on the heated pad. Two to three days later, the experiment was repeated, and CNO injection was administered just before the mouse was placed on the heated pad. The first 15 minutes following CNO injection were excluded from the analyses.

Methods section (page 18, lines 12-17)

For the experiments described in Fig. 5, immediately prior to the placement of the heated pad, the mice were given an IP saline injection, and when the experiment was repeated 3-4 days later, an IP injection with 10 mg/kg clozapine-N-oxide (CNO, Angene, London, UK). We analyzed the entire recording after saline injection, but excluded the first 15 minutes following CNO injection to allow for CNO absorption^{77,78}.

c. Given that the hM3Dq receptor is expressed pan-neuronally in the preoptic hypothalamus, it seems surprising that the same result is always produced. I would think that sometimes there would be more expression in wake-promoting preoptic neurons, depending on each animal where the virus is transduced exactly, and wakefulness is produced with CNO. There must be quite a lot of variability here? (glutamate neurons in the preoptic can produce wake when excited).

Answer: Indeed, the overall effect we observed with CNO was sleep-promoting across all tested mice (WT and DS). However, as also indicated in our answer to question 6a of this Reviewer, we did detect variability in the magnitude and the direction of the response to CNO with respect to the delta power contribution to wake and NREM at 36°C. While DS mice generally showed an enhanced delta contribution, this effect was not consistently seen in all of the animals, with one mouse even displaying a reduction in NREM delta power at 36°C (Fig. 5J green). We agree that such variability likely reflects differences in the specific cell populations of transduced in each mouse, and we have now included a statement discussing this potential source of variability in the revised Discussion (pages 12-13, lines 23-11).

In light of the intricate neuronal population of the hypothalamus, which includes both sleep- and wake-promoting neurons⁵⁷, pinpointing the cellular basis of these DS-associated sleep and thermoregulatory deficits remains challenging. This complexity is further illustrated by the presence of hypothalamic neuron types with unique properties, such as wake-promoting glutamatergic parvalbumin-positive neurons⁶⁰. Moreover, since *Scn1a* is expressed in both excitatory and inhibitory hypothalamic neurons³⁶, identifying the precise cellular mechanisms involved becomes even more difficult. However, the observation that hypothalamic neuronal CAV2-mediated SCN1A expression or chemogenetic stimulation restored the warmth-induced sleep-promoting response suggests that reduced neuronal excitability, resulting from *Scn1a* loss-of-function mutation, is a plausible mechanism for the observed DS-related deficits. Nevertheless, CAV2-mediated expression of SCN1A or hM3D is not restricted to a specific class of neurons. CAV2-SCN1A contains the neuronal-specific enolase (NSE) promoter, and CAV2-hM3D-IRES-mCitrine harbors the CMV (cytomegalovirus early enhancer) promoter. Both promoters lead to transgene expression in excitatory as well as inhibitory neurons²⁸. Moreover, with the effective retrograde transport properties characteristic of CAV2^{61,62}, injection of the CAV2-based vectors to the anterior hypothalamus also transduced neurons in regions that project to the hypothalamus. Thus, it is possible that the anterior hypothalamic injection of CAV2-SCN1A leads to a correction in the activity of multiple types of neurons in this region, as well as additional connecting networks that hub on the anterior hypothalamus,

that together contribute to the observed improvement in sleep and thermoregulation in DS mice.

And pages 14, lines 21-25

A limitation of the present study is that we did not directly assess how the effects of the DS *Scn1a* mutation and hypothalamic CAV2-mediated SCN1A expression, or alternatively chemogenetic stimulation, influence neuronal firing and excitability in the anterior hypothalamic region. Future studies addressing these questions could provide deeper mechanistic insight into the neuronal basis of sleep and thermoregulation deficits in DS.

Signed: William Wisden

Reviewer #2 (Remarks to the Author):

In this novel study, Dr. Rubinstein and colleagues shed light on the understudied yet crucial problems of thermal dysregulation and sleep disturbances in Dravet syndrome. Notably, they establish a connection between these two deficits in a mouse model of the disorder. Their findings reveal that Dravet mice have a lower baseline body temperature. Furthermore, through integrated EcoG, LFP, and temperature monitoring, they discovered that, unlike wild-type mice, which experience a temperature drop transitioning from wakefulness to NREM sleep, Dravet mice do not. Intriguingly, increasing body temperature induced sleep in wild-type mice but failed to do so in Dravet mice. Interestingly, when they restored Nav1.1 expression using a CAV2-Nav1.1 vector that they previously developed or induced neuronal activation using chemo-genetics in the anterior hypothalamus, warmth-induced sleep was rescued in the Dravet mice.

This work is novel, well-designed, and well-written and contributes to the field by highlighting the contribution of previously unexplored brain areas to the multifaceted phenotypes of Dravet syndrome. Publication of this important paper will be very valuable to the field.

Answer: Thank you for your helpful comments and suggestions, which we have addressed in the revised manuscript.

That said, there are some issues that need to be integrated or better characterized to maximize the informative power and the impact of the work.

1. The authors focused on the transition from wakefulness to NREM sleep to investigate the link between temperature dysregulation and sleep disturbances. However, to have a complete picture, it would be worth to study additionally the link between core body temperature alterations and the transition from NREM to REM phase. This is likely something not so difficult to do as they have already performed the recordings.

Answer: Unfortunately, in the relatively short recordings of one hour that we performed for these experiments, only very few epochs consistent with REM sleep were observed, and these were not consistently identified by both independent scorers. As unequivocal REM periods could therefore hardly be confirmed, this sleep stage was not suitable for reliable analysis. We are planning a follow-up study with extended 24 - 48-hour recordings, which will allow robust detection of REM sleep and enable us to address this important aspect in detail.

A statement on the lack of REM sleep data being available for analysis has now been included in the Methods section (page 17, lines 28-30).

The recordings were long enough, and the conditions were suitable for the vast majority of the mice (except for one DS mouse, which was excluded from the analysis) to reach NREM sleep, but not REM sleep.

2. The results of the two rescue experiments with injection of CAV2s expressing either Nav1.1 or hM3D are impressive. However, they could be more informative if the authors could better characterize the percentage and extent of neuronal transduction in the hypothalamus and interconnected brain areas, considering the retrograde properties of CAV2s.

Answer: We appreciate the suggestion and underscore the value of providing more information on the extent of neuronal transduction in the hypothalamus and interconnected brain areas. With endogenous Nav1.1 expression in WT and DS mice, the biodistribution could only be examined using a reporter protein.

The biodistribution of CAV2-SCN1A was examined using the control CAV2-GFP or CAV2-mCitrine, where the transgene is driven by the NSE promoter (Fadila et al. 2023: PMID 37192002). As shown in the newly added Supplementary Fig. 9a-g, mCitrine immunoreactivity was observed across multiple brain regions, consistent with the retrograde transport properties of CAV2 vectors. At the injection site (a), mCitrine-positive cells and fibers were observed in the preoptic region of the hypothalamus, as well as the anterior and lateral hypothalamus. Thalamic regions also contained numerous labeled cells. Some cortical labeling was observed in the agranular and piriform cortices, although the number of positive cells was relatively low. The amygdala displayed mCitrine-positive cells and fibers, mainly within the medial and basolateral nuclei.

Supplementary Fig. 9

(a-g). Biodistribution of CAV2-mCitrine (in which the mCitrine reporter is driven by the NSE promoter) following stereotaxic injection into the hypothalamic preoptic region. Due to CAV2's retrograde transport properties, injection into the hypothalamus resulted in expression in multiple regions of the hypothalamus, thalamus, amygdala, and some cortical regions. (a-g) mCitrine immunoreactivity was observed at the injection site (a), and mCitrine-positive cells and fibers were observed across the preoptic region, including the lateral and medial preoptic areas, as well as the anterior and lateral hypothalamus (a-h). Thalamic regions also contained numerous labeled cells (c-g). Some cortical labeling was observed in the agranular and piriform cortices (a-g), although the number of positive cells in those regions was relatively low. The amygdala displayed mCitrine-positive cells and fibers, mainly within the medial and basolateral nuclei. Scale bars: 1 mm.

These data are referred to in the Results section (pages 8-9, lines 30-4):

The biodistribution was examined in mice following CAV2-GFP injection (WT and DS, n = 3 for each genotype), which were also used for behavioral and electrophysiological examinations, as well as in a separate group of three WT mice injected with the control vector CAV2-mCitrine, which were only used for immunohistochemistry. The biodistribution analysis demonstrated expression in excitatory and inhibitory neurons in the hypothalamus and, due to CAV2's retrograde transport properties, also in interconnected brain regions (Fig. 4a, b, Supplementary Figs. 8 and 9).

Unfortunately, we did not obtain better expression images with CAV2-hM3D-IRES mCitrine in the brain. The amount of mCitrine, which is the ORF after the IRES, is very low (Mizuguchi et al. 2000; PMID:10933956). And, like numerous labs, we also attempted to use commercial antibodies against hM3D, but in our hands, the signal was weak and not specific. We are also

unaware of any study showing convincing hM3D immunoreactivity *in vivo*. Despite its limitations, the current pictures in Supplementary Fig. 9h and Fig. 5b show the injection site.

As shown in the new Supplementary Fig. 9i-n, using CAV2-hM3D-IRES-mCitrine in E1-transcomplementing cells (DK-E1), we observe mCitrine expression. Detection of mCitrine in DK-E1 cells seeded as a monolayer, with transduction of the majority of cells on the coverslip, is easier than detecting transduced neurons in a brain following *in vivo* expression. These data indicate that the vector quality (physical to infectious particle ratio) is excellent and that transgenes from CAV2-hM3D-IRES-mCitrine are expressed. Briefly, DK-E1 cells were incubated with 50 particles of CAV2-hM3D IRES-miCitrine/cell. 24 h post-incubation, mCitrine and hM3D immunofluorescence were examined.

In addition, we injected three WT mice, in the hypothalamus, with a reporter vector, CAV2-CMV-mCitrine, which harbors the CMV promoter driving mCitrine. CAV2-CMV-mCitrine injections led to robust mCitrine immunoreactivity in the hypothalamus and connecting regions.

These data were not added to the manuscript to avoid confusion, as the expression levels of mCitrine using CAV2-CMV-mCitrine reporter vector are much higher than those of CAV2-hM3D-IRES miCitrine (which enabled the easy detection of the vector in the brain). Yet it indicates the expected biodistribution in the hypothalamus and connecting regions, including thalamic and cortical areas.

(A-M) CAV2-CMV-mCitrine biodistribution in the brain. CAV2-CMV-mCitrine was injected into the hypothalamus (D was the site of injection), leading to expression in multiple connecting brain regions, including the hypothalamus (C, D: MnPO: median preoptic, vLPO: ventrolateral preoptic, MPO: medial preoptic); thalamus (J, K: DLG: dorsal lateral geniculate nucleus, VLG: ventral lateral geniculate nucleus; IGL: intergeniculate leaflet, Rt: Reticular nucleus of the thalamus, Hb: Habenula), some neurons in cortical regions (A-K: Cortical Regions: Pir: piriform cortex, Icj: Islands of Calleja, EnDo: Endopiriform nucleus, Ac: agranular cortex), and the amygdala (E-F). scale, 1mm.

3. In which neuronal cell types is Nav1.1 expressed in the hypothalamus? RNAscope or analysis of available single cell data set of adult mouse brain could help.

Answer: We added a combined RNAscope and immunofluorescence analysis in WT mice injected with CAV2-mCitrine. These data demonstrate the expression of mCitrine in both

excitatory and inhibitory neurons in the hypothalamus, as well as *Scn1a* mRNA expression in these two neuronal populations.

Moreover, we extracted the *Scn1a* mRNA expression from hypothalamic single-cell RNA-seq datasets (Steuernagel et al., 2022: PMID:36266547). These data are shown in Supplementary Fig. 8.

Supplementary Fig. 8

Scn1a expression in excitatory and inhibitory neurons in the hypothalamus

a-g. A Representative section of the hypothalamus of a WT mouse injected with CAV2-mCitrine. mCitrine immunofluorescence (green), and RNAScope (*Gad* in magenta, *Scn1a* in red) showing the expression of *Scn1a* mRNA in both excitatory and inhibitory neurons.

a. mCitrine and DAPI. b. *Gad* and DAPI, c. *Scn1a* and DAPI. d. mCitrine and *Gad*. e. *Scn1a* and mCitrine. f. *Scn1a*. g. mCitrine, *Gad* and *Scn1a*. Scale bar: 10 μ M

h. Single-cell RNA-sequencing data restructured based on the open-access hypothalamic single-cell RNA-seq data from Steuernagel et al. (Steuernagel et al., 2022), showing *Scn1a* transcript levels (nCount RNA) in multiple clusters of GABAergic (blue) and glutamatergic (green) neurons. The dashed lines represent the average expression level across all types of GABAergic neurons or all types of glutamatergic neurons.

Reviewer #2 (Remarks on code availability):

I don't have the experience to test the code.

Reviewer #3 (Remarks to the Author):

In this report, Fadila et al., seek to establish a connection between sleep homeostasis and thermoregulation in an SCN1A haplo-insufficient mouse, carrying a Dravet Syndrome-associated mutation. Mouse models of DS recapitulate several of features of disease including seizures, impaired sleep and poor thermoregulation. Changes in core body temperature coincide with sleep/wake state transitions and increasing ambient temperature can promote NREM sleep. The authors hypothesize that sleep and temperature homeostasis disturbances are linked in DS mice and seek to understand the mechanisms by which they are coordinated. There is potentially dual value in this approach as it stands to be informative regarding the relationship between sleep and thermoregulation in the disease but also could reveal principles mechanistically linking SCN1A function to sleep and temperature homeostasis. While this is an interesting manuscript with some very nice preliminary findings, there are many methodological concerns (detailed below) including a lack of appropriate controls, inappropriate statistical analyses, and a lack of experimental detail as well as very limited mechanistic insight rendering the manuscript of limited impact in its current form.

Thank you for your valuable comments and suggestions, which we have addressed in the revised manuscript.

Our specific concerns and suggestions are laid out below more or less organized by Figures.
Figure 1

// The authors claim that “DS mice increased their core more rapidly than WT mice” (line 90), however no quantification of the rate of change in core body temperature is provided.

Answer: We have removed references to the rate of core temperature increase, as this parameter was indeed not formally quantified.

When mice are returned to RT to recover, the genotypes have different core body temperatures. However, it is curious that the authors do not highlight that both genotypes ‘overshoot’ their temperature recovery, ending up at a temperature below the baseline core body temperature shown in 1B and 1C. Do both WT and DS mice have impaired recovery in this model? .

Answer: Thank you for this relevant observation and concern. It is correct that for both WT and DS mice, upon the return to RT, show a reduction in core body temperature that is lower than the baseline measurement at RT. This overshoot has been described before and is a part of an adaptive homeostatic response (Romanovsky, 2018, PMID: 30454596).

We added a clearer depiction of these results in Fig. 1e, showing that both WT and DS mice reached a lower body temperature, compared to baseline temperature. Yet, the absolute temperature in DS mice reached a lower body temperature than WT animals (p = 0.0009, Two-Way ANOVA followed by Holm-Sidak post-hoc analysis). We have adapted the description of these findings in the Results (page 5, lines 11-15) accordingly to address this aspect.

Yet, upon return to RT, DS mice returned to a lower temperature than that of WT mice (Fig. 1c - e, Supplementary Fig. 1). Of note, while both WT and DS mice reached a temperature that was lower compared to their initial core temperature, the overall temperature of DS mice 15 min after recovery at RT was lower again lower than that of WT animals (Fig. 1e).

We should further note that in the experiments, we included two cohorts of mice: one cohort of 10 WT and 6 DS mice was monitored for body temperature during both the heating and cooling phases, while the other cohort of 3 WT and 6 DS mice (all together 13 WT mice and 12 DS mice) was monitored only during the heating phase on the heat pad but not during the subsequent cooling phase after removal from the pad. These data were shown in the original Fig. We realized that combining data from the two different cohorts of mice in a single visualization (despite stating the number of mice in each panel in the legend) could be confusing, especially when addressing the overshoot aspect discussed above. To improve clarity, the datasets are now shown separately: the revised Fig. 1c-e that was explained above, thus only presents the mice that were followed across both the heating and cooling phases. New Supplementary Fig. 1b-c now depicts the graph that was shown in the original version of the manuscript, showing both cohorts, mice that were monitored only during the heating phase, as well as the mice shown in Fig. 1C-E that were monitored during both heating and cooling. The legends of both figures have been adapted accordingly.

The authors next evaluate nesting behavior in WT vs DS mice because nests can stabilize core body temperature during sleep. DS mice may not build nests as complex as their WT littermates however, the significance of these findings at this juncture (or anywhere for that matter in the manuscript) is unclear. While pre-sleep behaviors in mice are now well recognized. In the context of the current manuscript, a defect in these behaviors raises the possibility of important confounds: for example, perhaps DS mice defect in nesting contributes to a lack of movement that would raise body temperature prior to sleep. This possibility is not addressed.

//Moreover, the localization of these pre-sleep behaviors is still not well circumscribed in the brain.

Answer: We acknowledge that nest building is an innate but multifactorial behavior that can be affected by other DS-associated comorbidities, such as motor or social deficits. We also agree with the Reviewer that reduced pre-sleep nesting could, in principle, influence thermoregulation, as diminished movement or insulation may attenuate the typical pre-sleep increase in body temperature. This possibility provides an additional behavioral context in which the observed alterations in sleep-related temperature dysregulation in DS mice may arise.

Additionally, we concur that the neural substrates controlling pre-sleep behaviors like nesting remain poorly delineated, which limits definitive attribution of the observed phenotype to specific upstream mechanisms. Following an additional set of experiments conducted in response to question 2 from Reviewer #1, we determined that vector-mediated expression of SCN1A in the hypothalamus did not normalize nesting behavior. The findings are consistent with the DS-related *Scn1a* mutation affecting broader neural networks or additional comorbidities that contribute to pre-sleep behavior, while not excluding a contribution from hypothalamic dysfunction.

The following text has been added to the Results section (page 5, lines 25-27)

Nevertheless, nesting is a multifactorial behavior that can also be influenced by DS-associated comorbidities, including motor or social deficits, which may contribute to this phenotype.

And the Discussion (page 13, lines 17-24)

Nesting, which is linked to preoptic hypothalamic activity⁶³⁻⁶⁶, also remained abnormal in DS mice despite CAV2-SCN1A expression in the hypothalamus, indicating that, in the context of

DS, additional comorbidities or neuronal networks contribute to this phenotype, and impaired nesting may further contribute to sleep temperature dysregulation. Conversely, as CAV2-SCN1A injection into the hippocampus or thalamus corrected the epileptic and cognitive phenotypes of DS ²⁸, it remains to be determined whether CAV2-SCN1A injection to these sites could also correct the sleep and thermoregulation deficits described here.

// It is unclear how these findings fit with previous and subsequent data in the manuscript. How does nesting behavior related to core body temperature in DS mice, sleep and hypothalamic activity? If nesting behavior is abnormal in DS mice, could this argue that SCN1A mutations disrupt circuits involved in behaviors which regulate temperature? This line of questioning is distinct from hypothesis that SCN1A acts in key brain regions directly involved in regulating core body temperature (i.e. hypothalamus).that SCN1A acts in key brain regions directly involved in regulating core body temperature (i.e. hypothalamus).

Answer: The evaluation of nesting behavior was included as it relates to sleep preparation, thermoregulation, and neuronal activity in hypothalamic circuits known to be involved in the regulation of these processes. Prior studies have shown that nesting activates preoptic GABAergic and glutamatergic neurons (Tagawa et al., 2024, PMID:38594484), lesions to the preoptic area reduce nesting (Lee et al., 1999, PMID:10212050; Olazábal et al., 2002, PMID:11855898), and optogenetic activation of Vgat+ neurons in the preoptic area promotes nesting (Li et al., 2019, PMID:30459220). These observations, in our view, collectively support a link between nesting and hypothalamic activity relevant to thermoregulation and sleep onset. In our study, DS mice displayed a less complex nesting behavior compared to WT, together with altered sleep-wake parameters and temperature regulation. This convergence suggests potential interaction between these processes; however, we agree that the directionality and causality remain uncertain. A defect in nesting could, as indicated in the previous answer and in the adapted Discussion text, in principle, alter sleep thermoregulation, but this remains speculative. With respect to the role of hypothalamic SCN1A in regulating nesting behavior, as noted in the previous comment, vector-mediated expression of SCN1A in the hypothalamus did not normalize nesting behavior. While these findings do not exclude a contribution of hypothalamic dysfunction to impaired nesting behavior in DS, the observations are supportive of other neural networks or comorbidities contributing to nesting behavior pre-sleep. As indicated in the previous answer, statements concerning these aspects have been included in the Results and Discussion sections, as described above.

Figure 2

The authors next evaluate sleep and the change of core body temperature during wake -> NREM transitions in WT and DS mice. They find that while WT mice have a higher delta power in NREM compared to wake, DS mice do not exhibit any change in delta power. In addition, while WT mice experience a decrease in core body temperature at wake -> NREM transitions, there is no change in DS mouse core body temperature at transition states.

//There are major technical concerns with the sleep recordings. It is unclear from the methods section at what time of day the recordings were done and how the mice were housed during the recording session.

// Were animals protected from environmental disturbances? Were they acclimated to the recording chambers or temperature probes before recording sleep?

Answer: We regret that our initial description of the sleep recording conditions was not sufficiently detailed, which has caused uncertainty about the procedures. All recordings were performed during the light phase between 8am-6pm, with similar time-of-day variation across genotypes and experimental groups. No additional habituation steps were included before recording. All ECoG/LFP sleep recordings were performed in a separate room, in a padded Plexiglas box (for the sleep related temperature recordings) or 2-liter glass beaker (for the warmth-induced experiments) with wet food provided. All of the mice (except for one DS mouse, which was not included in the analyses as we focused on the transitions from wake to NREM) were observed to fall asleep under these conditions, indicating that the environment was adequately calm and suitable for natural sleep onset.

We have now clarified in the Methods section that (pages 17, lines 19-30):

Video-ECoG/LFP recordings were performed in a separate room during the light period (8 am-6 pm) for 2-5 h per mouse in a padded Plexiglas box (Fig. 2 and Fig. 4d) or a 2-liter beaker (Fig. 3, Fig. 4g-j, Fig. 5) with wet food provided.

...

The recordings were long enough, and the conditions were suitable for the vast majority of the mice (except for one DS mouse, which was excluded from the analysis) to reach NREM sleep, but not REM sleep.

// Additionally, the recovery period post-surgery/implantation seems relatively short (2-3 days).

Answer: We regret that the post-surgical recovery period is judged as too brief. Based on the mouse's behavioral and neurophysiological features in the recording setup, including wake and NREM cycles and related characteristic power properties of the ECoG/LFP signals in the WT animals, we consider a 2-3 day interval after ECoG surgery sufficient for full recovery before initiating recordings. Note that for recovery after LFP surgery, a 3-5 days recovery period was used, which we now added to the Method section (page 16, lines 18,20).

For the ECoG recordings in Fig. 2 we used mice at their 4th week of life. For the ECoG recordings in Fig. 4, the mice were 5-6 weeks old. The mice were given 2-3 days for recovery after surgery before recording.

...(page 16 line 31)

(for the LFP) The mice were given 3-5 days for recovery after surgery before recording.

In our experience, animals reliably regained normal activity, grooming, and weight within this period, and (as indicated above) show good LFP/ECoG quality, including sleep onset.

// The statistical analysis of the PSD is not appropriate. It is advised to conduct a 2-way ANOVA, comparing WT-WAKE, WT-NREM, DS-WAKE and DS-NREM. From Figure S1, it appears that the authors instead conducted t-tests to compare within genotypes only.

The authors state " In DS mice, the increase in delta power was smaller than in WT animals" (line 160). With the current statistical analysis, they cannot make such a claim (unless we missed this somewhere).

Answer: We regret that our initial description of the statistical analysis may have been unclear. In fact, all the PSD data were originally analyzed using a Two-Way repeated-measures ANOVA to compare wake vs NREM within each genotype (WT and DS, depicted in Fig. 2).

As suggested, in the revised version of the manuscript, we added a new Supplementary Fig. 4, which includes a comparison between genotypes.

Moreover, as advised, Two-Way ANOVA (considering that the delta power was measured in the same mouse during the wake and NREM stages), with Holm Sidak post-hoc analysis, is used throughout the Figures. This analysis demonstrates that the difference in % delta did not reach statistical significance when comparing WT vs DS. We have included these details in the revised Methods (page 20, lines 7-10)

Data are reported as mean \pm standard error (SE). All statistical tests were carried out via GraphPad Prism. The details of the specific test and the p-values are listed in the Figure legend, with the full description listed in Supplementary Table 1. Differences were considered significant at $p < 0.05$.

Results sections (page 6, lines 17-22)

The PSD of wakefulness was calculated two minutes prior to the transition to NREM, and the PSD of NREM was measured on the first minute into NREM. In WT mice, a comparison of the PSDs of these states showed that the transition to NREM was accompanied by a clear increase in ECoG delta power. Conversely, while in WT mice this translated in the contribution of delta power to the overall ECoG being larger during NREM sleep than waking, in the DS mice, the delta ratio was similar during wake and NREM (Fig. 2c-d).

In addition, we updated the corresponding figure legends to explicitly state which test was used.

// As a general comment for all sleep recordings done in the report, there are often poor representative ECoG & EMG recording traces. Figure 2A, for example, displays ECoG and EMG traces in 'wake' and 'NREM' which are entirely indistinguishable from each other.

Answer: We understand this Reviewer's observation and regret that the presentation of the LFP/ECoG and EMG traces did not provide a convincing impression of the wake-NREM differentiation. This limitation is especially apparent in the short-duration examples shown in Fig. 2 and Fig. 4, which display only a few minutes of recording, as we are focusing on the transition from wakefulness to NREM sleep.

To address this, we changed the example in Fig. 2a, selecting an example that better exemplifies the transition, which also shows changes in the EMG activity (that were not too prominent in the previous example, in which the detection of the wake-NREM transition relied mostly on delta power and ECoG). As described in the Methods, a virtual channel derived from the ECoG/LFP signal displaying delta power was used to support the identification of NREM epochs, as well as the EMG channel. An example of this channel was added to Figs. 2 and 4. We would further like to note that the EMG channel was used to provide additional support for classifying NREM sleep, particularly in DS mice, which exhibit lower delta activity increases during this stage, while accounting for residual muscle activity that can persist during NREM sleep. The poor quality of NREM sleep in DS mice, reflected by the less pronounced increase in delta power compared to wakefulness, poses an inherent challenge for accurate sleep classification in this model. To address this, two independent investigators manually identified sleep transitions, and only transitions on which both agreed were included in the analysis.

To visualize the wake-sleep identifications also in the longer-duration(1 h) recordings from Figs. 3, 4, and 5, we included hypnogram depictions of the wake and NREM epochs, leaving out the transition periods for clarity. Supplementary Fig. 3 contains a detailed depiction of all identified stages, including the transitions.

We have clarified our vigilance state analysis approach in the revised Methods section (pages 18-19, lines 25-4):

Wakefulness was identified by desynchronized ECoG/LFP patterns and relatively high levels of EMG activity. Non-rapid-eye-movement (NREM) sleep was marked by high-amplitude, slow wave patterns accompanied by low EMG activity. Virtual channels derived from the ECoG/LFP signal, particularly delta power (0.5-4 Hz), were used to support NREM stage identification. However, in DS mice, as the increased delta activity during NREM is diminished, the EMG signal contributed to the classification. To distinguish rapid-eye-movement (REM) sleep from wakefulness, a theta (4-8 Hz) to delta power ratio of >2.5 , along with low EMG activity relative to NREM, was applied in the classification. Periods where wakefulness and NREM sleep alternated within 20 seconds were classified as transition states. Example recordings with hypnograms including all stages are shown Supplementary Fig. 3, whereas the hypnograms in Figs. 3-5 are shown without indication of these transitions for improved clarity. Transition periods, which did not allow for classification, were excluded from analysis.

Figure 3

// the authors argue that WT mice exhibit an increase in delta power after heating, but it should be addressed whether this effect is driven by one mouse, which seems to have a significant change (3D).

// The authors also employ a 'sleepiness scale' to evaluate the relative sleepiness of WT and DS mice after heating. However, it is unclear as to why the development or use of this 'sleepiness scale' is required. We question whether the use of EMG signal as a marker for sleepiness, in itself, is arbitrary and inappropriate. For example, a mouse with a severed spinal cord or with absence of spinal motor neurons may have different/absent EMG signals but no change in "sleepiness" but this scale would render a "sleepier" mouse simply as a function of immobility. Without validation or generalizability of such a scale, the data from it is hard to understand.

//Furthermore, this scale not used in Figure 2 recordings -- why not? For this and subsequent figures (3, 4 and 5) the authors also no longer separate a PSD analysis from wake vs NREM and it is unclear why (in line 227). It seems that the main effect of PSD delta power was in NREM stages (figure 2) and they may be missing an opportunity to have stronger effects.

Answer: We thank the reviewer for these thoughtful comments. We acknowledge that the concept and presentation of the "sleepiness scale" raised confusion, as was also noted by Reviewer #1. We fully agree that EMG activity alone does not uniquely define sleep or sleepiness, and that reduced muscle tone may occur in contexts other than sleep.

In light of this and the other Reviewer's concerns, we have removed the "sleepiness scale" from the manuscript. Instead, as advised, we now rely on % NREM and ECoG/LFP delta contribution during wake and NREM power to more objectively reflect sleep-wake dynamics.

We have added for all data-sets: (1) the percentage of time the mice spent in NREM sleep, and (2) the relative contribution of delta frequencies to the PSD during both wake (2 min before NREM onset) and NREM sections (1 min into NREM), as was shown in Fig. 2. The additional analyses of relative delta power during the wake states and NREM are shown in Fig. 3e, 3f, Fig. 3k, 3l, Fig. 4e, and Fig. 5f, 5j.

The data from WT mice, in all of the cohorts and the experimental conditions, show a warm-induced increase in % NREM, whereas in DS mice, a warm-induced increase in NREM is missing (Figs. 3-5).

Moreover, in WT mice, there is an increase in the contribution of delta power during as the mouse transitions to NREM sleep (Fig. 3e), whereas in DS, none of the examined cohorts demonstrated this increase (untreated mice with sleep at RT, Fig. 2d, 2j, untreated mice at RT or 36 °C, Fig. 3f, or DS mice injected with CAV2-hM4D-mCitrine without CNO injection, Fig. 5j blue).

Nevertheless, in WT mice in Fig. 3k at 36 °C and in Fig.5e at 36 °C or on the 2nd recording day at RT, the contribution of delta to NREM was not statistically different from that of wake. In Fig. 3k, we suspect that this is related to the relatively small number of mice examined (4 mice), with the possibility that the experimental procedure of viral injection or LFP surgery, in one of the mice, may have caused disruption in delta-driven brain regions in one of these mice (although also in this mouse the % NREM was increased by warmth).

In Fig 5f, 5 mice were tested, but since there are more conditions, reaching statistical significance is more difficult.

This was added to the Results section (page 8, lines 15-18)

In WT mice injected with CAV2-SCN1A, similar to untreated WT mice, the change to 36°C increased the duration of NREM sleep (Fig. 3j, k). However, a significant increase in the delta contribution at NREM was only observed at RT, likely due to the relatively low number of examined mice in this control cohort (Fig. 3k).

We hope that these updated analyses provide a clearer, validated assessment of sleep-wake behavior without relying on the previously proposed scale.

Figure 4

The authors test whether expression of SCN1A in the anterior hypothalamus can restore warmth-mediated sleep in DS mice. They argue that CAV2-mediated SCN1A expression in DS mice restores the increased NREM time and delta power induced by heating.

// However, this experiment is lacking appropriate control (empty virus not “untreated”) and statistical analyses. While it appears the authors had an ‘untreated’ group (showed in supplements) they do not include these groups in their statistical analyses.

Answer: Thanks. To address this concern, we performed an additional set of experiments in another cohort of mice, in which we conducted ECoG recordings to compare warmth-induced sleep in WT and DS mice that were injected with either a control vector (CAV-GFP) or CAV-SCN1A.

The Methods section has been updated accordingly (page 17, lines 14-15):

WT and DS mice, at the age of P22-26, were injected with CAV2-NSE-SCN1A, CAV2-NSE-GFP, CAV2-NSE-mCitrine (CAV2-SCN1A, CAV2-GFP or CAV2-mCitrine²⁸), or CAV2-hM3D-IRES-mCitrine (Plateforme de Vectorologie de Montpellier, IGMM Montpellier, France).

In the Results section (pages 8-9, lines 28-28), we added that:

To directly control for the effects of the hypothalamic CAV2 injections, a separate cohort of WT and DS mice was injected with a control vector containing a reporter gene CAV2-GFP, or with CAV2-SCN1A. The biodistribution was examined in mice following CAV2-GFP injection (WT and DS, n = 3 for each genotype), which were also used for behavioral and electrophysiological examinations, as well as in a separate group of three WT mice injected

with the control vector CAV2-mCitrine, which were only used for immunohistochemistry. The biodistribution analysis demonstrated expression in excitatory and inhibitory neurons in the hypothalamus and, due to CAV2's retrograde transport properties, also in interconnected brain regions (Fig. 4a, b, Supplementary Figs. 8 and 9).

In the cohort of CAV2-GFP or CAV2-SCN1A injected mice, we also evaluated nest-building behavior. Seven to ten days after vector injection, CAV2-SCN1A did not rescue the impaired nest-building abilities of DS mice (Supplementary Fig. 10). Next, ECoG electrodes were implanted, and the mice were allowed to recover for 2-3 days before we proceeded with ~4 h ECoG and temperature recordings at RT. Also in this cohort of mice, CAV2-SCN1A expression in the hypothalamus restored the increase in delta ratio during the transition to NREM sleep, which was absent in DS mice injected with the control vector (CAV2-GFP, Fig. 4a-e). Of note, these analyses were conducted using cortical ECoG rather than the hypothalamic LFP recordings used for the CAV2-SCN1A experiments shown in Fig. 3, confirming that the correction of sleep-related oscillatory activity by CAV2-SCN1A is also reflected at the cortical level. Core temperature measurements were included in this new cohort of mice to test whether hypothalamic SCN1A expression may also correct the drop in body temperature at NREM onset. However, in contrast to the data from naïve untreated mice shown in Fig. 2, no significant temperature decrease was detected after NREM onset in either WT or DS mice injected hypothalamically with either CAV2-GFP or CAV2-SCN1A. The discrepancy between these results and those presented in Fig. 2 may be attributable to the different experimental procedures, as the mice depicted in Fig. 2 underwent only one surgical process, for electrode implantation (ECoG at the age of 4 weeks, or LFP at the age of 5 weeks), while the mice shown in Fig. 4 underwent two surgeries, one for vector injection at 4 weeks and a second one for ECoG implantation at 5 weeks.

The warmth-induced somnogenesis experiment was also performed in this cohort of mice, 1-4 days after the ECoG recordings at RT, and confirmed that NREM sleep was promoted in DS mice injected with CAV2-SCN1A, whereas in DS mice injected with the control CAV2-GFP vector, the switch to warm temperature did not induce NREM sleep (Fig. 4f-j).

Fig. 4: CAV2-SCN1A restores NREM sleep delta ratio and warmth-induced somnogenesis in DS mice

- (a-b) The control vector CAV2-mCitrine was injected into the hypothalamus. LPO: Lateral preoptic area, MPO: medial preoptic area, vLPO: ventrolateral Preoptic nucleus: anteroventral periventricular nucleus. Scale bar: 1 mm.
- (c) mCitrine expression in the vLPO. Scale bar 10 μ M
- (d) Examples showing the ECoG signal, EMG, sonogram, and the virtual delta depicting delta (0.5-4 Hz) power in a WT: CAV2-GFP, WT: CAV2-SCN1A, DS: CAV2-GFP and DS: CAV2-SCN1A during a transition from wakefulness to NREM sleep.
- (e) About 2 weeks after viral injection, the mice were recorded for 4 h at RT. The relative contribution of delta power during the analyzed wake and NREM periods (WT: CAV2-GFP, n = 7. WT: CAV2-SCN1A, n = 6. DS: CAV2-GFP, n = 6. DS: CAV2-SCN1A, n = 5. $p = 0.023$, Two-way ANOVA. The results of Holm-Sidak post hoc analyses are depicted on the graph).
- (f) The effect of warm temperature was tested in the same mice 4-6 days later, at this stage, the mice were recorded for 1h at RT, and then on a pad heated to 36°C. The percent of time the mice spent in NREM sleep at each temperature, is depicted (WT: CAV2-GFP, n = 5. WT: CAV2-SCN1A, n = 6. DS: CAV2-GFP, n = 4. DS: CAV2-SCN1A, n = 5. $p < 0.0001$, two-way ANOVA. The results of Holm-Sidak post hoc analyses are depicted on the graph).
- (g-j) Examples of ECoG, EMG, sonogram and hypnogram of WT: CAV2-GFP (g), WT: CAV2-SCN1A (h), DS: CAV2-GFP (i), DS: CAV2-SCN1A (j).

// And again, the authors fail to statistically compare WT and DS mice against each other. They could do this by statistically comparing the change in delta ratio from RT -> 36°C in each injection x genotype group.

Answer: We added the statistical comparison between genotypes for the %NREM related to the data in Figs. 3 and 4 to the new Supplementary Fig. 6.

Supplementary Fig. 6

A full statistical comparison between different experimental groups shown in Figs. 3, 4.

- (a). The experimental paradigm, used for examining the effect of warmth on sleep promotion.
- (b-c). The same data shown in Figs. 3d and 3j are replotted here to highlight genotype differences. (2-way ANOVA. The results of Holm-Sidak post hoc analyses are depicted on the graph).
- (d). The data shown in Figs. 4f are shown again here with the inclusion of the full statistical analysis to highlight also genotype differences. (2-way ANOVA. The results of Holm-Sidak post hoc analyses are depicted on the graph).

* $p < 0.05$; ** $p < 0.01$; *** $p < 0.001$

Figure 5

The authors test whether chemogenetic activation of the anterior hypothalamus using activating DREADDS can rescue warmth-mediated sleep in DS mice.

//The rationale for the experiment is only partially clear. What is known about the activity of the neurons in anterior hypothalamus in DS mice? Is the activity in the haplo-insufficient model lower or higher than WT mice? In which cells? Testing this would support the DREADD experiments.

Answer: Thank you for these comments. Little is known about the activity of the hypothalamus in DS, mainly because cellular studies are challenging in this region due to its diverse neuronal population. However, as indicated in our response to a related question from Reviewer #2 (i.e., question 3, pages 14-15 of this rebuttal), RNAseq data, which are extracted from Steuernagel et al.(2022, PMID: 36266547) and replotted here, and our RNAscope analyses of three WT mice demonstrated comparable expression of *Scn1a* in both excitatory and inhibitory neurons. These data are presented in the new Supplementary Fig. 8 that we included on page 15 of this rebuttal.

The DS model used here harbors a missense *Scn1a* mutation; it is not a haplo-insufficient model. However, like other DS models, the *Scn1a* mutation causes a loss of function effect, and electrophysiological recordings in the hippocampus of these mice demonstrated reduced firing on inhibitory neurons, reduced hippocampal inhibition, and altered firing and network connections between excitatory neurons (Almog et al., 2022, PMID: 33271326, 2021, PMID: 35370551; Fadila et al., 2023, PMID: 37192002; Layer et al., 2023, BioRxiv). Nevertheless, additional studies are needed to directly examine the activity of specific classes of hypothalamic neurons in DS.

Thus, the rationale for testing chemogenetic activation of the hypothalamus is that DS-associated *Scn1a* loss-of-function mutations are expected to reduce neuronal excitability. This is in line with the observation of reduced overall spectral power in DS, as seen in our hypothalamic LFP recordings (Supplementary Fig. 4). The findings presented in Fig. 5 indeed support the view that chemogenetic activation, similar to exogenous SCN1A expression, improved warmth-induced NREM in DS mice.

To address that, we added to the Results section an expanded explanation for our reasoning: (page 10, lines 4-12)

The prevailing view is that DS-associated *Scn1a* mutations lead to reduced neuronal excitability due to impaired Nav1.1 channel function^{34,35}. The hypothalamus comprises a heterogeneous population of excitatory and inhibitory neurons, both expressing comparable levels of *Scn1a* mRNA (Supplementary Fig. 8)³⁶. If the primary effect of CAV2-SCN1A is to enhance the activity of transduced neurons, we wondered whether artificially increasing neuronal excitability, through chemogenetic activation, could mimic the effect of exogenous Nav1.1 expression. Specifically, we tested whether chemogenetic activation of hypothalamic neurons could restore temperature-dependent sleep induction, similar to the correction observed following CAV2-SCN1A injections.

We expanded the Discussion on these findings to underscore the complementary insight of the CAV2-SCN1A vs DREADD experiments (pages 12-13, lines 23-11).

In light of the intricate neuronal population of the hypothalamus, which includes both sleep- and wake-promoting neurons⁵⁷, pinpointing the cellular basis of these DS-associated sleep and thermoregulatory deficits remains challenging. This complexity is further illustrated by the presence of hypothalamic neuron types with unique properties, such as wake-promoting

glutamatergic parvalbumin-positive neurons⁶⁰. Moreover, since *Scn1a* is expressed in both excitatory and inhibitory hypothalamic neurons³⁶, identifying the precise cellular mechanisms involved becomes even more difficult. However, the observation that hypothalamic neuronal CAV2-mediated SCN1A expression or chemogenetic stimulation restored the warmth-induced sleep-promoting response suggests that reduced neuronal excitability, resulting from *Scn1a* loss-of-function mutation, is a plausible mechanism for the observed DS-related deficits. Nevertheless, CAV2-mediated expression of SCN1A or hM3D is not restricted to a specific class of neurons. CAV2-SCN1A contains the neuronal-specific enolase (NSE) promoter, and CAV2-hM3D-IRES-mCitrine harbors the CMV (cytomegalovirus early enhancer) promoter. Both promoters lead to transgene expression in excitatory as well as inhibitory neurons²⁸. Moreover, with the effective retrograde transport properties characteristic of CAV2^{61,62}, injection of the CAV2-based vectors to the anterior hypothalamus also transduced neurons in regions that project to the hypothalamus. Thus, it is possible that the anterior hypothalamic injection of CAV2-SCN1A leads to a correction in the activity of multiple types of neurons in this region, as well as additional connecting networks that hub on the anterior hypothalamus, that together contribute to the observed improvement in sleep and thermoregulation in DS mice.

We also added to the Discussion that cellular electrophysiological studies within this region are challenging, and future studies are needed to address this important question (page 14, lines 21-25).

A limitation of the present study is that we did not directly assess how the effects of the DS *Scn1a* mutation and hypothalamic CAV2-mediated SCN1A expression, or alternatively chemogenetic stimulation, influence neuronal firing and excitability in the anterior hypothalamic region. Future studies addressing these questions could provide deeper mechanistic insight into the neuronal basis of sleep and thermoregulation deficits in DS.

// There is again a lack of proper control (empty vector) and appropriate approach to statistical analysis. Including a CNO only control would be ideal as the effects observed could be CNO or hypothalamus-mediated.

Answer: To directly test the possibility that CNO may cause sleepiness in mice without DREADD expression, we examined the effect of 10 mg/kg CNO on locomotor activity in four awake WT mice assessed for a period of 10 minutes in the open field starting 15 minutes after CNO injection. These data were added to the new Supplementary Fig. 10; the locomotor activity was similar following CNO (10 mg/kg) and saline injections.

We added the following statements to:

The Methods section (page 18, lines 17-20):

Locomotor activity of WT mice, without DREADD expression, was not affected by 10 mg/kg CNO. Specifically, 15 min after CNO injection, locomotor activity of the mice was monitored for 10 minutes in an open field of 50 by 50 cm (Supplementary Fig. 12).

And the Results section (page 10, lines 20-23):

This dosage of CNO did not alter locomotor activity, as assessed in an open field test in WT mice without DREADD expression (Supplementary Fig. 12), indicating the absence of a nonspecific sedation effect at this dose³⁷.

Supplementary Fig. 12

CNO administered at 10 mg/kg had no effect on the locomotor activity of WT mice. WT mice were placed in a 50 x 50 cm open field arena 15 min after CNO injection, and their locomotor activity within the arena was recorded for 10 min. Four days later, the same mice were injected with saline, and their locomotor activity was recorded again 15 min after saline injection over 10 min. The similar distance covered under both conditions indicates that CNO at this dose does not have a non-specific sedative effect.

And the Discussion section (page 14, lines 21-25).

A limitation of the present study is that we did not directly assess how the effects of the DS *Scn1a* mutation and hypothalamic CAV2-mediated SCN1A expression, or alternatively chemogenetic stimulation, influence neuronal firing and excitability in the anterior hypothalamic region. Future studies addressing these questions could provide deeper mechanistic insight into the neuronal basis of sleep and thermoregulation deficits in DS.

// there is no confirmation that CNO administration actually had an effect on anterior hypothalamus activity.

Answer: As noted also in our answer to a question from this Reviewer on Fig. 5 (as well as to a related question from Reviewer #1), cellular electrophysiological studies of the hypothalamus are extremely challenging. Therefore, the effect of CNO was examined at the neuronal network and behavioral level only, focusing on the ability of CNO to promote NREM sleep in a warm environment in DS mice. As noted in our earlier answers, we addressed this limitation in the Discussion.

Other major concerns

- Lack of description about the haplo-insufficient DS mouse model

Answer: We added to the Introduction previous studies of this animal model (page 3, lines 11-14).

Among the many existing models, DS mice harboring the missense A1783V *Scn1a* mutation exhibit severe epilepsy and various DS-associated behavioral phenotypes, including spatial memory and motor deficits^{12,18-21}. Sleep properties or thermoregulation have not been investigated in this DS model.

- Varied ages for the mouse studies

Answer: The use of two mouse age groups was dictated by technical considerations. DS mice at 4 weeks of age are at heightened risk for seizures and premature death. Therefore, for

more invasive procedures such as hypothalamic depth LFP electrode implantation, and for experiments involving hypothalamic viral injections followed by depth LFP or ECoG surgery, we performed the LFP and ECoG surgery in 5-week-old mice, which have a significantly lower risk for surgical complications and SUDEP. Additionally, sufficient time for post-surgical recovery and viral expression necessitated the use of an older cohort for the viral transfection studies. Crucially, sleep and thermoregulation deficits were observed in both younger (4-week-old) and older (5-week-old) DS mice, demonstrating that the phenomenon is not confined to a single age group. This consistency across cohorts, in our view, supports the validity of our observations.

We included an adapted statement concerning the rationale for using 5-week-old mice for experiments involving depth LFP recordings in the Results section (page 7, lines 2-3)

Of note, as these depth surgeries are more invasive, we used 5-week-old mice.

and included in the Discussion (page 14, lines 18-20)

Nonetheless, sleep and thermoregulation deficits were observed in both juvenile (4-week-old) and older (5-6-week-old) DS mice, demonstrating that the phenomenon is not confined to a single age group or disease stage.

- baseline temperature difference is gone in older animals

Answer: The absence of a baseline temperature difference in older DS mice is not considered a concern but rather reflects an age-related change in the disease phenotype. We hypothesize that this age-related normalization of core body temperature reflects a reduction in seizure burden rather than an experimental confound.

This observation and a discussion on the interpretation has now been included into the Discussion (page 14, lines 4-20):

Another observation we made was that juvenile DS mice, at the beginning of their fourth week of life, showed a lower baseline core temperature compared to WT mice. This difference could be attributed to altered metabolic homeostasis described in DS ⁷², involving dysregulation of energy metabolism and mitochondrial function, or to the severe epilepsy typical at this age. Both factors may contribute to the reduced thermoregulatory stability observed across sleep-wake transitions. In the DS model used here, all the mice experience spontaneous seizures during their fourth week of life ^{12,27}. The lower baseline temperature may therefore be related to post-ictal hypothermia, which has been reported in both human epilepsy patients ⁷³ and DS mice up to several hours after a seizure ⁷⁴. In older DS mice at 5-6 weeks of age, when spontaneous seizures become less frequent ^{12,27}, the baseline temperature difference between DS and WT mice was less pronounced, which is consistent with a reduced seizure burden, as well as a reduction in metabolic stress. Although we did not assess metabolic parameters in the present study, this will be an important direction for future investigations to better define the contribution of metabolic mechanisms to thermoregulatory regulation in DS.

- many seizures in the younger animals, do any of them die?

Answer: We did not observe any deaths during our recording, but about 50% of our DS mice were found dead in the cage following electrode implantation or viral injection. This is the rate we see with untreated DS mice (Fadila et al, 2020, PMID:32865826, Fadila et al. 2023, PMID: 37192002).

- Lack of investigation and discussion about potential metabolic contributions to differences in basal and sleep-state transition associated thermoregulation

Answer: We appreciate the Reviewer's suggestion to consider potential metabolic contributions to the observed thermoregulatory differences. Although metabolic parameters were not directly investigated in the present study, previous reports have documented altered metabolic homeostasis in DS models, including changes in energy expenditure and mitochondrial function. Such alterations, together with the pronounced seizure burden in juvenile DS mice, may contribute to the reduced baseline core temperature and impaired thermoregulatory functions during sleep-wake transitions. This aspect has been incorporated into the revised Discussion (page 14, lines 4-20);, where we also note that a direct assessment of metabolic involvement would be an important objective for future studies.

Another observation we made was that juvenile DS mice, at the beginning of their fourth week of life, showed a lower baseline core temperature compared to WT mice. This difference could be attributed to altered metabolic homeostasis described in DS ⁷², involving dysregulation of energy metabolism and mitochondrial function, or to the severe epilepsy typical at this age. Both factors may contribute to the reduced thermoregulatory stability observed across sleep-wake transitions. In the DS model used here, all the mice experience spontaneous seizures during their fourth week of life ^{12,27}. The lower baseline temperature may therefore be related to post-ictal hypothermia, which has been reported in both human epilepsy patients ⁷³ and DS mice up to several hours after a seizure ⁷⁴. In older DS mice at 5-6 weeks of age, when spontaneous seizures become less frequent ^{12,27}, the baseline temperature difference between DS and WT mice was less pronounced, which is consistent with a reduced seizure burden, as well as a reduction in metabolic stress. Although we did not assess metabolic parameters in the present study, this will be an important direction for future investigations to better define the contribution of metabolic mechanisms to thermoregulatory regulation in DS.

- No investigation into the activity of the hypothalamus in DS mice, this could link hypothalamic involvement to the temperature-sleep transitions

Answer: We agree with the Reviewer that the lack of hypothalamic neuronal recordings is a limitation of the current study and have addressed this in the revised Discussion (page 14, lines 21-25), as also noted above.

A limitation of the present study is that we did not directly assess how the effects of the DS *Scn1a* mutation and hypothalamic CAV2-mediated SCN1A expression, or alternatively chemogenetic stimulation, influence neuronal firing and excitability in the anterior hypothalamic region. Future studies addressing these questions could provide deeper mechanistic insight into the neuronal basis of sleep and thermoregulation deficits in DS.

- There is also occasional inconsistency in the PSD findings – sometimes the main significant effect of the experimental variable is seen in delta power and sometimes it is in delta ratio, but there is a lack of discussion about this.

Answer: Following your advice, we have now consistently examined the contribution of delta frequencies to wake and NREM, as well as the percentage of NREM sleep, during RT and warm environment. These analyses are now presented instead of the delta power, which was previously shown in the text and was derived from sections that combined wake and NREM, resulting in less consistent results.

In the results shown in Figs. 2-5, we consistently see an increase in % NREM in warm temperatures in WT mice and a lack of increase in % NREM sleep in DS mice.

Reviewer #4 (Remarks to the Author):
